# SERE: Similarity-based Expert Re-routing for Efficient Batch Decoding in MoE Models

**Juntong Wu[1,2,*], Jialiang Cheng[1,*,†,✉], Fuyu Lv[1], Ou Dan[1], Li Yuan[2,✉]**
[1] Taobao & Tmall Group of Alibaba
[2] Shenzhen Graduate School, Peking University
**Correspondence:** jichen.cjl@alibaba-inc.com, yuanli-ece@pku.edu.cn

## Abstract

Mixture-of-Experts (MoE) architectures employ sparse activation to deliver faster training and inference with higher accuracy than dense LLMs. However, in production serving, MoE models require batch inference to optimize hardware efficiency, which may cause excessive expert activation and thus slow the memory-bound decoding stage. To address the fundamental tension between batch decoding and expert sparsity, we present **SERE**, a **S**imilarity-based **E**xpert **R**e-routing method for **E**fficient batch decoding in MoE models. SERE dynamically reduces the number of active experts in an input-aware manner by re-routing tokens from secondary experts to their most similar primary counterparts. It also leverages similarity patterns to identify and preserve critical experts, thereby preventing capability loss. Notably, SERE avoids static expert pruning or merging, instead enabling dynamic expert skipping based on batch-level expert redundancy. Additionally, we provide an efficient custom CUDA kernel for SERE, enabling plug-and-play use in vLLM with only a single-line code change.[1] Extensive experiments on various complex reasoning benchmarks demonstrate that SERE achieves up to $2.0\times$ speedup with minimal quality loss, providing a practical solution for cost-efficient and latency-sensitive large-scale MoE deployment.

## 1 Introduction

Large Language Models (LLMs) have shown remarkable performance across various applications. Recently, the Mixture-of-Experts (MoE) paradigm has emerged as a leading framework for scaling LLMs (Yang et al., 2025a; Liu et al., 2024b; Touvron et al., 2023; Jiang et al., 2024). Unlike dense LLMs that activate the entire feed-forward network (FFN) for every token, an MoE layer consists of multiple lightweight FFN experts, where a learnable router assigns each token to a small subset. By maintaining low per-token computation, sparse activation enables the model to incorporate numerous specialized experts, scaling its capacity while preserving training and inference efficiency.

Despite the theoretical efficiency of MoE architectures, their practical gains are often limited by a mismatch between selective activation and batched inference (Kwon et al., 2023; Agrawal et al., 2024; Gupta et al., 2024). In real-world services, multiple user requests are batched to improve hardware utilization (Kwon et al., 2023). However, tokens within a batch often require different experts, leading to a total number of activated experts far above the per-token budget (Agrawal et al., 2024; Yun et al., 2024). As depicted in Figure 1, even with strict limits (e.g., 8 out

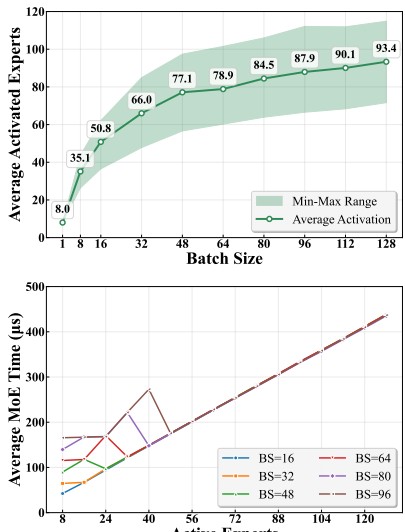

Figure 1: Larger batches activate more experts. With a fixed batch size, more experts increase decoding time.

---

* Equal contribution    † Project Lead    ✉ Corresponding author
[1] Code implementation of SERE can be found in `https://github.com/JL-Cheng/SERE`.

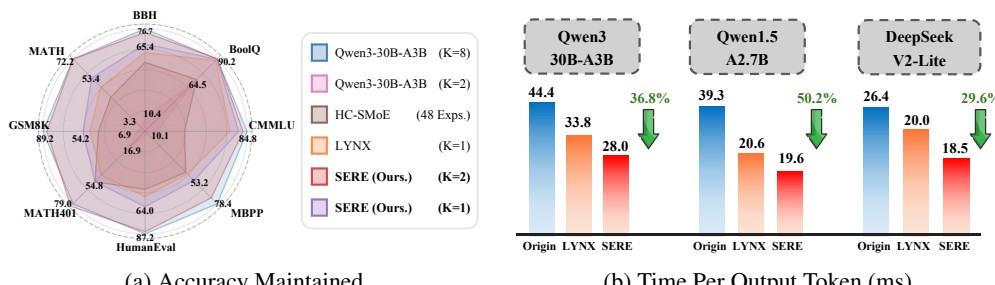

(a) Accuracy Maintained    (b) Time Per Output Token (ms)

Figure 2: Visualizations of SERE's Performance. (a) Across all tasks, SERE ($K$=2) exhibits negligible performance loss, while SERE ($K$=1) still outperforms all baselines. (b) SERE significantly reduces batch decoding time, achieving up to $2\times$ acceleration.

of 128 in Qwen3-30B-A3B (Yang et al., 2025a)), a moderately diverse batch can still activate a majority of the experts simultaneously. Moreover, the training-time load-balancing objectives further increase the expert diversity within a batch (Lepikhin et al., 2021; Liu et al., 2024b). This issue is particularly acute during decoding (Yun et al., 2024), where sequential token generation makes the process memory-bandwidth-bound. As also can be seen from Figure 1, activating excessive experts during decoding raises communication and memory-access overhead and thus increases latency. Addressing the conflict between batched inference and sparse expert activation is therefore crucial for unlocking the practical scalability of MoE architectures (Zoph et al., 2022; Liu et al., 2024b).

To address the problem mentioned above, various expert-reduction methods are proposed, which can generally be classified into static model compression and dynamic expert skipping. Static methods typically remove or merge experts in a fixed, pre-defined manner (Yang et al., 2024; Liu et al., 2024c; Chen et al., 2025; Ai et al., 2025). While these methods can efficiently reduce the memory footprint, they typically involve significant computational costs, rely on task-specific insights, and might reduce the model's capacity and ability to generalize. Dynamic methods modify expert activation at runtime based on token-level signals (Zhong et al., 2024; Huang et al., 2024; Lu et al., 2024; Gupta et al., 2024; Yang et al., 2025b). These methods depend solely on router scores, overlook intrinsic expert characteristics, and often require extra training or threshold tuning. Moreover, their complex token-by-token operations or modification of the decoding process hinder integration with high-performance inference frameworks, such as vLLM (Kwon et al., 2023), which limits their practicality in large-scale deployment.

Starting with these observations, we propose SERE, a **S**imilarity-based **E**xpert **R**e-routing method for **E**fficient batch decoding in MoE models. SERE is motivated by three key observations. First, many experts within an MoE layer exhibit high functional similarity. Therefore, SERE re-routes tokens from a subset of experts to their most similar counterparts, reducing the number of active experts with minimal capacity loss. Second, a small set of high-ranked primary experts dominate gating weights and output contributions, whereas secondary experts contribute little. SERE retains all primary experts and only re-routes secondary ones, thereby preserving dominant contributors while minimizing redundancy. Third, certain critical experts are highly dissimilar to others and specialize in unique input patterns. SERE preserves these experts to prevent capability degradation during re-routing. In summary, SERE employs a dynamic, input-aware strategy that jointly considers token characteristics and inter-expert similarity, skipping more experts when redundancy is high and fewer when diversity is essential for accuracy. The expert similarity matrix is pre-computed once from a general calibration set, requiring no retraining or task-specific tuning. For deployment, we implement an efficient custom CUDA kernel for SERE that can be seamlessly integrated into the widely used vLLM framework (Kwon et al., 2023), enabling plug-and-play use with only a single-line code change.

The contributions of our work are summarized as follows:

1. We propose SERE, a similarity-based expert re-routing method for accelerating batch decoding in MoEs. SERE significantly reduces the number of active experts while maintaining model performance, enabling faster decoding.

2. We develop an efficient, plug-and-play CUDA kernel for SERE that works with various MoE models and can be easily integrated into the vLLM framework (Kwon et al., 2023).

3. We perform extensive experiments on multiple state-of-the-art MoE models. (Bai et al., 2023; Liu et al., 2024a; Yang et al., 2025a). As shown in Figure 2, SERE achieves up to $2.0\times$ speedup with minimal impact on output quality.

## 2 RELATED WORK

Recent work on expert reduction can be mainly divided into two categories: static model compression and dynamic expert skipping.

**Static Model Compression** methods leverage redundancy among experts to perform pruning or merging operations. For example, MoE-$I^2$ (Yang et al., 2024) reduces the size of MoE models via a two-stage process of inter-expert pruning and intra-expert low-rank decomposition. EEP (Liu et al., 2024c) employs an evolutionary search that prunes experts and merges their knowledge into the remaining subsets. HC-SMoE (Chen et al., 2025) applies hierarchical clustering based on expert similarity to iteratively merge similar experts. Other approaches, such as DeRS (Zhang et al., 2025a), $D^2$-MoE (Gu et al., 2025), and ResMoE (Ai et al., 2025), represent experts with shared weights augmented by low-rank residuals. While effective in reducing model size, these methods often incur high computation costs, rely heavily on calibration data and task-specific priors, and risk reducing the model's capacity and generalization ability due to decreased expert diversity.

**Dynamic Expert Skipping** aims to reduce the number of activated experts during inference dynamically. For instance, Top-$p$ routing (Huang et al., 2024) selects experts dynamically based on the confidence scores for each input. AdaMoE (Zhong et al., 2024) and MoE++ (Jin et al., 2025) enable token-adaptive routing via introducing null experts. Yang et al. proposes a layer-wise and fine-grained top-$k$ reduction strategy to improve inference efficiency. NAEE (Lu et al., 2024) skips less critical experts via token-wise analysis of router weights, and LYNX (Gupta et al., 2024) employs batch-aware confidence estimation to filter out less relevant experts for unimportant tokens. While effective in reducing computation, these methods often require extra training, operate at coarse granularity, and overlook intrinsic expert characteristics by relying solely on router scores. Their per-token operations also incur overhead and are challenging to integrate with high-performance inference frameworks, limiting their practical benefits in large-scale deployment.

## 3 METHOD

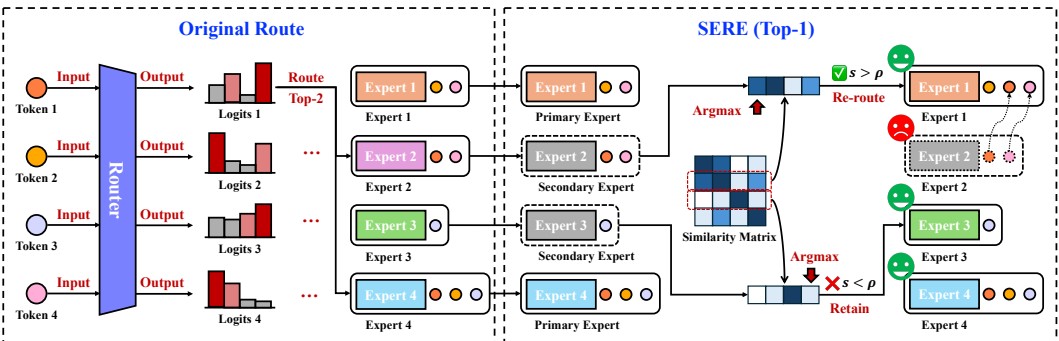

Figure 3: Illustration of SERE with 4 tokens and 4 experts as example. Tokens are first routed to top-2 experts. SERE preserves the primary experts (1 and 4) and re-routes the secondary experts (2 and 3). As a result, Expert 2 is replaced by Expert 1, while Expert 3 remains active as its similarity to all active experts falls below the threshold.

To accelerate batched decoding in MoE models, we propose SERE, a dynamic, input-aware expert skipping method. As illustrated in Figure 3, SERE preserves the primary experts for all tokens as well as the critical experts within each layer, and re-routes tokens from secondary experts to their most similar retained counterparts. This dynamic strategy achieves substantial decoding speedups while maintaining model performance. In the remainder of this section, we introduce the design motivations and technical components of SERE. We begin with **expert similarity estimation** (Sec.

3.1), then describe the **similarity-based dynamic re-routing mechanism** (Sec. 3.2), and finally present the **implementation of a high-performance CUDA kernel** for integration into large-scale inference frameworks (Sec. 3.3).

## 3.1 EXPERT SIMILARITY ESTIMATION

### 3.1.1 SIMILARITY MATRIX COMPUTATION

We adopt a data-driven approach to measure expert similarity in MoE models. Consider an MoE model with $L$ layers, where each layer $l$ contains $M$ experts $\{\mathbf{E}_1^{(l)}, \ldots, \mathbf{E}_M^{(l)}\}$. Using a calibration dataset $\mathcal{D}_{\text{calib}}$, we process $N$ batches and aggregate the results to obtain robust similarity estimates. For each batch $i \in [1, N]$, let $\mathbf{X}_i^{(0)}$ denote the input embeddings. In each layer $l$, expert activations are obtained as $\mathbf{A}_{i,j}^{(l)} = \mathbf{E}_j^{(l)} \left( \mathbf{X}_i^{(l-1)} \right)$, after which pairwise similarities are computed via a predefined similarity function $\text{Sim}(\cdot, \cdot)$:

$$\mathbf{S}_{p,q}^{(l)} += \text{Sim} \left( \mathbf{A}_{i,p}^{(l)}, \mathbf{A}_{i,q}^{(l)} \right), \quad 1 \leq p, q \leq M. \tag{1}$$

Common choices of $\text{Sim}(\cdot, \cdot)$ include Cosine Similarity, Frobenius norm, and centered kernel alignment (CKA) (Kornblith et al., 2019). More details can be found in Appendix A.2.

After all $N$ iterations, the accumulated similarity matrices are normalized to obtain the average layer-wise similarity: $\mathbf{S}^{(l)} = \mathbf{S}^{(l)}/N$. The resulting set $\{\mathbf{S}^{(l)}\}_{l=1}^L$ provides a quantitative view of the similarity relationships between experts within the same layer. High similarity values indicate potentially redundant experts, while low values reflect diverse expert specialization. The pseudocode is provided in Algorithm 1 in Appendix A.4.

### 3.1.2 SIMILARITY MATRIX INSIGHTS

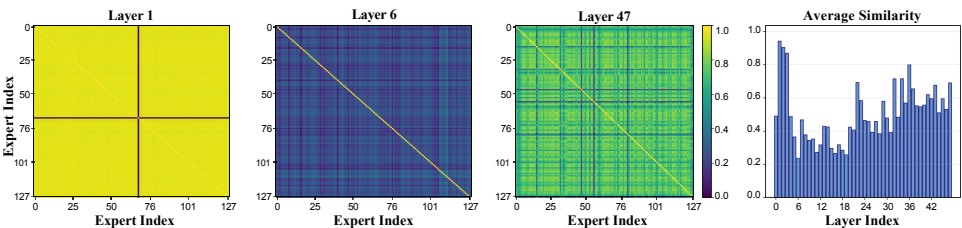

Figure 4: Visualization of the expert similarity matrices and the average expert similarity across all layers in Qwen3-30B-A3B (Yang et al., 2025a).

We computed the expert similarity matrices for all layers of the Qwen3-30B-A3B model (Yang et al., 2025a), with representative heatmaps and layer-wise average statistics shown in Fig. 4. The results reveal three notable patterns. First, within each layer, groups of experts exhibit consistently high pairwise similarity, indicating functional redundancy. Second, similarity patterns vary substantially across layers — Layer-1 has the highest average similarity, with nearly all pairs above $0.9$, while Layer-6 has the lowest average similarity, with most pairs below $0.4$. Third, every layer contains *critical* experts whose similarity to all others is exceptionally low, as indicated by heatmaps that display distinct horizontal and vertical stripes. Even in Layer 1, Expert 92 stands out as a critical expert, with a similarity of less than $0.1$ to all others. These observations illustrate the balance between redundancy and specialization in MoE architectures, highlighting that certain experts contribute uniquely to model capacity while others may provide overlapping functionality. More visualization results of expert similarity matrices for different MoE models are provided in Appendix C.3. These results demonstrate that high expert similarity is common in MoE models, regardless of whether upcycling initialization (Komatsuzaki et al., 2023) is employed.

> **Key Insights:**
> **1. Layer-wise redundancy:** Within each layer, groups of experts exhibit high pairwise similarity.
> **2. Cross-layer variation:** Average expert similarity varies substantially across layers.
> **3. Critical experts:** Each layer contains critical experts with uniformly low similarity to all others.

## 3.2 SIMILARITY-BASED EXPERT RE-ROUTING MECHANISM

### 3.2.1 DESIGN MOTIVATION

To accelerate batch decoding in MoE models by reducing the number of active experts, two key questions arise: (1) *Which active experts should be skipped?* and (2) *How should they be handled?*

For the first question, analysis of router weights distribution (Fig. 5) reveals that top-ranked (*primary*) experts dominate output activations and should therefore be retained, whereas low-ranked (*secondary*) experts contribute less and are natural skip candidates. To address the second question, we leverage insights from Sec. 3.1.2. Because layers contain groups of highly similar experts, tokens from a skipped secondary expert can be re-routed to its most similar retained primary expert, thus mitigating disruption to output activations. However, the analysis also identifies critical experts whose removal would degrade performance. We therefore introduce a similarity threshold that ensures such critical experts are always retained.

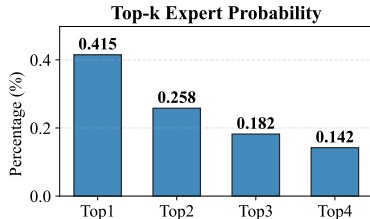

Figure 5: Weights Distribution

### 3.2.2 RE-ROUTING PROCESS

Building upon the observations and motivation, we now present our SERE method in detail. Let $\mathbf{R}^{(l)}(\cdot)$ denote the router function in layer $l$ of the MoE model. For a token $t \in \mathcal{T}$, $R^{(l)}(t) = (\mathbf{E}_{r_1}^{(l)}, \mathbf{E}_{r_2}^{(l)}, \ldots, \mathbf{E}_{r_K}^{(l)})$ is the ordered list of $K$ experts selected for $t$ by descending router weight, and $r_k \in \{1, \ldots, M\}$ denotes the index of the $k$-th ranked expert.

**Step 1: Primary expert selection.** We identify the primary expert set in layer $l$ as the union of the Top-$S$ experts over all tokens in the current batch:

$$\mathcal{E}_p^{(l)} = \bigcup_{\mathcal{T}} \left\{ \mathbf{E}_{r_k}^{(l)} \mid 1 \leq k \leq S \right\}. \tag{2}$$

Here, $S \in [1, K)$ is a hyperparameter controlling the size of the primary expert set. Smaller $S$ leads to fewer activated experts and higher acceleration, but may degrade quality. Experts in $\mathcal{E}_p^{(l)}$ are considered important and are always retained.

**Step 2: Similarity-based re-routing for secondary experts.** For each secondary expert $\mathbf{E}_u^{(l)} \in \left( \bigcup_{t \in \mathcal{T}} R^{(l)}(t) \right) \setminus \mathcal{E}_p^{(l)}$, we use the similarity matrix $\mathbf{S}^{(l)}$ to find its most similar primary expert:

$$\text{sim}_u^* = \max_{E_v^{(l)} \in \mathcal{E}_p^{(l)}} \mathbf{S}_{u,v}^{(l)}, \quad v_u^* = \arg\max_{\mathbf{E}_v^{(l)} \in \mathcal{E}_p^{(l)}} \mathbf{S}_{u,v}^{(l)}. \tag{3}$$

If $\text{sim}_u^* \geq \rho$, where $\rho \in [0, 1]$ is a similarity threshold, we re-route all tokens originally assigned to $\mathbf{E}_u^{(l)}$ to the most similar primary expert $\mathbf{E}_{v_u^*}^{(l)}$. If $\text{sim}_u^* < \rho$, $\mathbf{E}_u^{(l)}$ is determined as a critical expert and preserved to avoid unsafe substitutions. It should be noted that the re-routing process does not modify the router weights. The formulaic expression is as follows:

$$\forall t_j : \ \mathbf{E}_u^{(l)} \in R^{(l)}(t_j) \ \wedge \ \text{sim}_u^* \geq \rho \implies \mathbf{E}_u^{(l)} \leftarrow \mathbf{E}_{v_u^*}^{(l)}. \tag{4}$$

**Step 3: Final execution.** After re-routing, the final active expert set in layer $l$ is:

$$\mathcal{E}_{\text{final}}^{(l)} = \mathcal{E}_p^{(l)} \ \cup \ \{\mathbf{E}_u^{(l)} \mid \text{sim}_u^* < \rho\}, \tag{5}$$

which contains all primary experts and any preserved critical secondary experts. The MoE layer then utilizes this updated token-to-expert mapping to produce the output activations.

### 3.3 High-performance kernel implementation

We further develop a high-performance, hardware-friendly, and plug-and-play CUDA kernel for SERE. The implementation is model-agnostic, compatible with a wide range of MoE architectures, and can be integrated seamlessly into the vLLM framework (Kwon et al., 2023) without requiring modifications to its core execution pipeline. The pseudocode is outlined in Algorithm 2 in Appendix.

In practice, this CUDA-accelerated SERE achieves substantial speedups in batch decoding while preserving model accuracy, making it readily deployable in both research and production environments. Besides, enabling SERE requires only a single additional line of code, ensuring effortless adoption in existing MoE inference pipelines.

## 4 Experiments

### 4.1 Experiment Settings

**Models** We evaluate SERE on three representative MoE models: Qwen1.5-MoE-A2.7B-Chat (Bai et al., 2023), DeepSeekV2-Lite (Liu et al., 2024b), and Qwen3-30B-A3B (Yang et al., 2025a).

**Baselines** We compare SERE against several SOTA methods, including HC-SMoE (Chen et al., 2025), Top-K reduction (Yang et al., 2025b), and LYNX (Gupta et al., 2024). All baselines are implemented using official code or reproduced in strict accordance with the original papers to ensure a fair comparison.

**Benchmarks** For accuracy evaluation, we use reasoning tasks from OpenCompass (Contributors, 2023) across three domains: **Exam** (CMMLU (Li et al., 2024), BoolQ (Clark et al., 2019), BBH (Suzgun et al., 2023)), **Math** (Math (Hendrycks et al., 2021), GSM8K (Cobbe et al., 2021), Math_401 (Yuan et al., 2023)), and **Code** (HumanEval (Chen et al., 2021), MBPP (Austin et al., 2021)). CoT mode is used for CMMLU and BoolQ. For acceleration evaluation, we measure *Time per Output Token* (TPOT) under varying *Queries per Second* (QPS) using vLLM (Kwon et al., 2023), with each model deployed on a single GPU. Input/output lengths are fixed at 128/32 tokens.

**Hyper-Parameters** We use the Frobenius norm as the similarity metric and FineWeb-Edu (Lozhkov et al., 2024) (400 sequences×128 tokens) as the calibration dataset. For expert merging methods, pruning rates are chosen to match the TPOT of expert skipping methods for a fair comparison. All experiments are conducted on NVIDIA H20 GPUs.

For more detailed settings, please refer to Appendix B.

### 4.2 Accuracy Comparison

We comprehensively evaluate SERE and competitive baselines on the aforementioned models and benchmarks, Table 1, 2, and 3 present both accuracy and per-token decoding latency (TPOT).

| Methods \ Tasks | Exam | | | Math | | | Code | | Avg. | TPOT |
|---|---|---|---|---|---|---|---|---|---|---|
| | cmmlu | boolq | bbh | math | gsm8k | math$_{401}$ | heval | mbpp | (Acc. ↑) | (ms. ↓) |
| Qwen1.5-A2.7B $_{top4}$ | 69.58 | 80.46 | 34.97 | 14.38 | 51.86 | 60.60 | 45.73 | 30.60 | 48.52 | 17.29 |
| Qwen1.5-A2.7B $_{top2}$ | 66.69 | 75.87 | 32.26 | 12.92 | 44.28 | 51.36 | 32.02 | 27.40 | 42.85 | **13.53** |
| HC-SMoE $_{40\ experts}$ | 45.11 | 74.95 | 29.01 | 4.26 | 27.67 | 42.64 | 4.88 | 1.80 | 28.79 | 14.20 |
| LYNX $_{top2}$ | 42.57 | 78.62 | 23.59 | 9.56 | 29.57 | 34.41 | 8.54 | 7.40 | 29.28 | 14.49 |
| SERE $_{top2;\ \rho=0.0}$ | 68.12 | **80.15** | 33.16 | 14.06 | 50.19 | **58.35** | **46.95** | 26.20 | 47.15 | 13.83 |
| SERE $_{top2;\ \rho=0.3}$ | **68.49** | 79.97 | **34.61** | **14.66** | 51.63 | **58.35** | 42.68 | **27.60** | **47.25** | 13.93 |
| Qwen1.5-A2.7B $_{top1}$ | 45.12 | 48.35 | 29.47 | 5.24 | 26.16 | 46.13 | 15.85 | 14.80 | 28.89 | **11.47** |
| HC-SMoE $_{30\ experts}$ | 7.93 | 34.19 | 29.14 | 1.72 | 8.95 | 29.18 | 0.61 | 0.00 | 13.97 | 13.30 |
| LYNX $_{top1}$ | 16.60 | 77.68 | 15.10 | 0.68 | 2.12 | 10.22 | 0.00 | 0.20 | 15.33 | 12.95 |
| SERE $_{top1;\ \rho=0.0}$ | 60.09 | **79.85** | 32.71 | 7.58 | 33.36 | 52.12 | **17.07** | 20.20 | 37.87 | 12.13 |
| SERE $_{top1;\ \rho=0.3}$ | **65.83** | 78.69 | **33.62** | **9.74** | **39.88** | **53.12** | **17.07** | **20.60** | **39.82** | 12.95 |

Table 1: OpenCompass and TPOT (QPS=16) results on Qwen1.5-MoE-A2.7B. **Bold** for the best.

| Methods \ Tasks | Exam | | | Math | | | Code | | Avg. (Acc. ↑) | TPOT (ms. ↓) |
|---|---|---|---|---|---|---|---|---|---|---|
| | cmmlu | boolq | bbh | math | gsm8k | math$_{401}$ | heval | mbpp | | |
| DeepSeekV2-Lite $_{top6}$ | 53.34 | 82.39 | 49.37 | 23.82 | 59.14 | 70.32 | 54.27 | 45.40 | 54.76 | 26.35 |
| DeepSeekV2-Lite $_{top2}$ | 36.91 | 73.67 | 42.51 | 15.90 | 52.39 | 65.84 | 40.85 | 34.80 | 45.36 | **19.51** |
| HC-SMoE $_{48\ experts}$ | 39.74 | 80.70 | 41.97 | 9.16 | 47.92 | 45.14 | 10.98 | 7.00 | 35.33 | 22.36 |
| LYNX $_{top2}$ | 16.32 | 68.62 | 19.68 | 9.06 | 31.92 | 33.67 | 10.37 | 2.40 | 24.01 | 22.07 |
| SERE $_{top2;\ \rho=0.0}$ | **53.13** | 82.11 | 48.67 | 23.04 | **61.03** | **71.07** | 56.10 | 45.80 | 55.12 | 21.60 |
| SERE $_{top2;\ \rho=0.3}$ | 53.04 | 82.02 | **49.11** | **23.80** | 60.50 | 69.83 | **58.54** | **47.00** | **55.48** | 23.12 |
| DeepSeekV2-Lite $_{top1}$ | 19.41 | 58.90 | 33.81 | 2.56 | 17.82 | 48.88 | 7.93 | 7.60 | 24.61 | **18.02** |
| HC-SMoE $_{32\ experts}$ | 26.51 | 63.06 | 33.48 | 0.94 | 6.29 | 13.97 | 0.00 | 0.80 | 18.13 | 20.28 |
| LYNX $_{top1}$ | 2.16 | 49.91 | 3.96 | 0.14 | 1.29 | 2.00 | 0.00 | 0.00 | 7.43 | 20.00 |
| SERE $_{top1;\ \rho=0.0}$ | **53.81** | 82.11 | 48.69 | **23.74** | 58.53 | **72.32** | 57.93 | 45.40 | 55.32 | 18.54 |
| SERE $_{top1;\ \rho=0.3}$ | 53.49 | **82.63** | **48.90** | 22.94 | **59.36** | 71.32 | **59.15** | **47.20** | **55.62** | 20.59 |

Table 2: OpenCompass and TPOT (QPS=16) results on DeepSeekV2-Lite. **Bold** for the best.

| Methods \ Tasks | Exam | | | Math | | | Code | | Avg. (Acc. ↑) | TPOT (ms. ↓) |
|---|---|---|---|---|---|---|---|---|---|---|
| | cmmlu | boolq | bbh | math | gsm8k | math$_{401}$ | heval | mbpp | | |
| Qwen3-30B-A3B $_{top8}$ | 84.88 | 90.21 | 76.70 | 72.28 | 89.23 | 79.05 | 87.20 | 78.40 | 82.24 | 44.40 |
| Qwen3-30B-A3B $_{top2}$ | 10.01 | 60.52 | 10.48 | 3.38 | 6.97 | 16.96 | 3.66 | 2.40 | 14.30 | **30.97** |
| HC-SMoE $_{80\ experts}$ | 45.62 | 83.94 | 65.11 | 59.86 | 79.23 | 64.84 | **86.59** | 70.20 | 69.42 | 39.14 |
| LYNX $_{top2}$ | 81.36 | 90.12 | 72.27 | 69.10 | 80.44 | 76.81 | 84.15 | **73.40** | 78.46 | 38.21 |
| SERE $_{top2;\ \rho=0.0}$ | 81.24 | 89.79 | 71.33 | 70.22 | 82.41 | 80.80 | 82.93 | 63.80 | 77.82 | 32.12 |
| SERE $_{top2;\ \rho=0.5}$ | **81.51** | **90.37** | **74.15** | **72.06** | **85.97** | **81.55** | 85.37 | 72.00 | **80.37** | 32.82 |
| Qwen3-30B-A3B $_{top1}$ | 0.00 | 61.68 | 4.89 | 0.08 | 0.91 | 1.25 | 0.00 | 0.00 | 8.60 | **27.28** |
| HC-SMoE $_{48\ experts}$ | 32.78 | 64.53 | 51.66 | 34.36 | 40.79 | **54.86** | 49.39 | 44.20 | 46.57 | 33.45 |
| LYNX $_{top1}$ | 70.76 | 88.26 | 59.08 | 44.28 | 48.37 | 47.88 | 55.49 | 46.00 | 57.52 | 33.38 |
| SERE $_{top1;\ \rho=0.0}$ | 60.53 | 85.08 | 57.64 | 46.98 | 52.08 | 52.12 | 32.32 | 31.40 | 52.27 | 28.04 |
| SERE $_{top1;\ \rho=0.5}$ | **77.89** | **89.76** | **65.45** | **53.40** | **54.28** | **54.86** | **64.02** | **53.20** | **64.11** | 33.10 |

Table 3: OpenCompass and TPOT (QPS=16) results on Qwen3-30B-A3B. **Bold** for the best.

SERE consistently achieves the best trade-off between accuracy and inference efficiency. With aggressive expert skipping (e.g., Top-2), SERE maintains over $97\%$ of the original model's accuracy across all tasks, while reducing decoding latency by up to $1.6\times$ on Qwen3 and $1.4\times$ on Qwen1.5 and DeepSeekV2. In contrast, direct Top-$K$ reduction yields the lowest latency but causes severe performance degradation (up to $90\%$ accuracy drop), indicating a significant loss of model capacity.

HC-SMoE and LYNX achieve competitive performance on Qwen3 but show significant accuracy drops on Qwen1.5 and DeepSeekV2, particularly for math and code tasks. This may stem from architectural differences: Qwen3 contains more fine-grained and redundant experts, allowing greater tolerance to merging or skipping, whereas Qwen1.5 and DeepSeekV2 have fewer, more specialized experts and are thus more sensitive to expert selection. Methodologically, HC-SMoE's static merging reduces expert diversity, while LYNX ignores expert characteristics, thereby both impairing reasoning capability. In contrast, SERE incorporates both inter-expert similarity and the preservation of critical experts into its dynamic skipping strategy, removing redundancy while safeguarding essential capacity, thereby delivering consistently superior performance across all models and tasks.

Furthermore, we can observe that SERE performs well even without preserving critical experts ($\rho = 0$), while preservation ($\rho > 0$) brings further accuracy gains with negligible latency. The similarity threshold provides fine-grained control over the trade-off between capability and speed.

## 4.3 ACCELERATION COMPARISON

In this section, we compare the acceleration performance of different methods across multiple models and QPS settings. As shown in Figure 6, SERE consistently achieves substantial reductions in decoding latency under all evaluated QPS conditions. For Qwen3 and DeepSeekV2, SERE yields a **1.2**$\times$ to **1.6**$\times$ speedup, while for Qwen1.5, the acceleration ratio reaches up to **2.0**$\times$ when QPS$=$ 24, with almost no performance loss (See Section 4.2). Moreover, the CUDA-implemented SERE de-

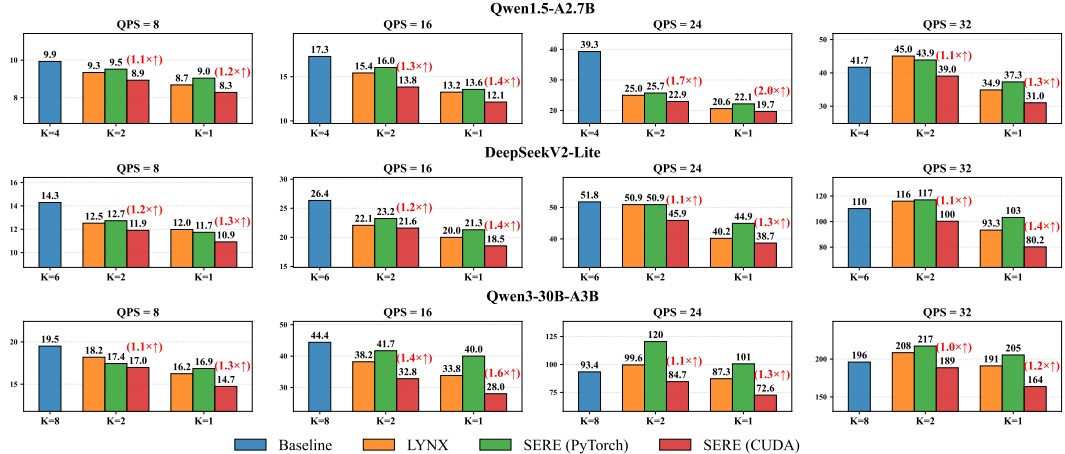

Figure 6: Batched Inference Latency between different methods in different QPS and Top-K.

livers approximately **1.5×** speedup over the PyTorch version. Besides, as shown in Figure 7c, the additional re-routing overhead is negligible relative to expert computation and remains stable across batch sizes. These results confirm the efficiency of the custom CUDA kernel.

We further analyze the variation in the average activated expert count under different Top-$K$ settings. As shown in Fig. 7a, the count grows logarithmically with batch size for all Top-$K$ values, and larger $K$ consistently leads to more activations. The results indicate that the primary activated experts for different tokens are highly concentrated, which explains why SERE achieves significant acceleration. Furthermore, Fig. 7b shows that the inter-layer differences in activated expert count become more pronounced as $K$ increases, highlighting the importance of dynamic expert skipping, where more aggressive skipping is applied to layers with higher activations.

We also examine how SERE behaves in the prefill stage, with details presented in Appendix C.2.

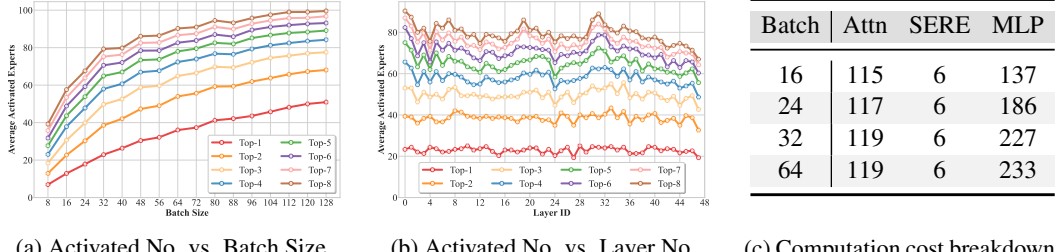

(a) Activated No. vs. Batch Size.    (b) Activated No. vs. Layer No.    (c) Computation cost breakdown.

Figure 7: (a)&(b) Average activated expert count of Qwen3-30B-A3B under different Top-$K$: variation with batch size and across layers (batch size=32). (c) Computational cost breakdown ($\mu$s) of key MoE operations for Qwen3-30B-A3B at varying batch sizes.

## 4.4 ABLATION STUDY

**Ablation on Similarity Threshold** We conduct experiments under both Top-1 and Top-2 settings, varying the threshold from $0.0$ to $1.0$, where $\rho = 1.0$ corresponds to the original model without any expert skipping. The resulting speedup and average accuracy for the three models are shown in Figure 8. Up to a point, increasing the threshold improves accuracy while adding only negligible decoding latency. Beyond that point, accuracy continues to rise, but the speedup drops sharply, indicating that too many active experts are being retained. In practice, the threshold at this inflection point offers a good balance between accuracy and latency. For example, in the case of Qwen3-30B-A3B, a threshold of $0.5$ achieves this balance. We also notice that DeepSeekV2 maintains relatively stable performance across different settings, whereas Qwen3 and Qwen1.5 exhibit notable perfor-

mance fluctuations. This finding highlights the substantial architectural and functional differences among different MoE models. More analysis on threshold can be found in Appendix C.1.

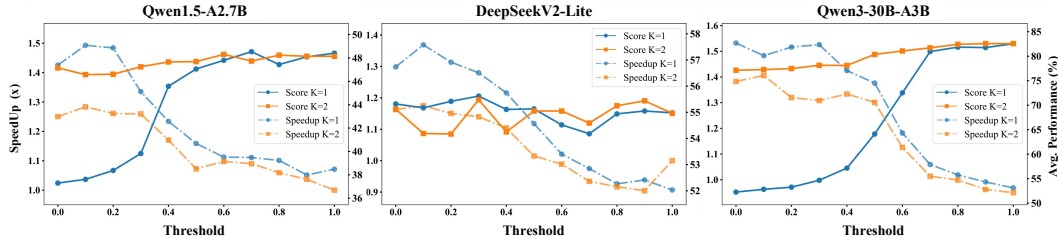

Figure 8: Speedup and Performance (Acc) under different similarity thresholds across models.

**Ablation on Similarity Matrix Computation** We investigate how different similarity metrics, parameter-based similarity measures, calibration datasets, and calibration data volumes used to compute the expert similarity matrix can potentially affect the overall performance and stability of SERE across downstream tasks. For similarity metrics, we compare Frobenius similarity, cosine similarity, and CKA-based similarity (Kornblith et al., 2019). For the data-free, parameter-based similarity computation methods, we follow Zhang et al. (2025b) and adopt two strategies for combining expert parameters: (1) **Concat** method that directly concatenates the three weight matrices $\{\theta_1, \theta_2, \theta_3\}$, and (2) **Logic** method that constructs a composite weight as $\theta_3(\theta_1 \cdot \theta_2)$. For calibration datasets, we use general datasets including FineWeb-Edu (Lozhkov et al., 2024), C4 (Raffel et al., 2020), and WIKI (Merity et al., 2017), together with domain-specific datasets derived from specific domains (Exam, Math, Code) and the mixed domains (OpenCompass). For calibration data volume, we experiment with three configurations: $200 \times 64$, $400 \times 128$, and $800 \times 256$. Additional experimental details can be found in Appendix A.3 and Appendix B.4.

Table 4 shows that SERE is highly robust to different similarity metrics, and Frobenius provides the fastest calibration. By comparing Table 4 with Table 5, we observe that the parameter-based similarity computation methods perform significantly worse than the activation-based methods. This suggests that capturing functional similarity through dynamic activations is more effective than computing similarity based on static expert parameters.

Table 6 reports the performance of SERE under different calibration datasets and calibration data volumes. When $K = 2$, the performance remains highly consistent across different calibration datasets and data volumes, indicating that SERE is robust with respect to both the type and the scale of calibration data. We also find that even when calibrated with domain-specific data, SERE maintains strong performance on other domains, which suggests that the similarity matrix captures transferable expert relationships rather than overfitting to a particular domain. Furthermore, when $K = 1$, domain-specific calibration provides slightly better results than general calibration, indicating that in high skipping rate settings, using domain-specific calibration data can further improve the performance of SERE.

Balancing calibration efficiency, effectiveness, and universality, we choose Frobenius Similarity and the FineWeb-Edu calibration dataset as our final implementation.

| Method | K=1 | | | | K=2 | | | | Time |
|---|---|---|---|---|---|---|---|---|---|
| | Exam | Math | Code | AVG | Exam | Math | Code | AVG | Cost (s.) |
| Frobenius | 57.55 | 31.02 | 18.64 | 37.87 | 60.48 | 40.87 | 36.58 | 47.15 | 28 |
| Cosine | 57.90 | 26.57 | 17.16 | 35.96 | 60.49 | 38.57 | 33.62 | 45.55 | 75 |
| CKA-RBF | 58.39 | 31.29 | 19.98 | 38.62 | 60.94 | 40.84 | 34.74 | 46.85 | 16064 |
| CKA-Poly | 57.50 | 29.59 | 20.26 | 37.72 | 60.63 | 40.68 | 32.59 | 46.13 | 13459 |
| CKA-Linear | 58.12 | 29.77 | 19.46 | 37.83 | 60.57 | 40.31 | 35.18 | 46.62 | 541 |
| **Mean±Std** | $\mathbf{57.89}_{\pm 0.36}$ | $\mathbf{29.65}_{\pm 1.84}$ | $\mathbf{19.10}_{\pm 1.20}$ | $\mathbf{37.60}_{\pm 0.94}$ | $\mathbf{60.62}_{\pm 0.18}$ | $\mathbf{40.25}_{\pm 0.93}$ | $\mathbf{34.54}_{\pm 1.54}$ | $\mathbf{46.46}_{\pm 0.60}$ | / |

Table 4: Comparisons across different similarity metrics on Qwen1.5-MoE-A2.7B.

**Ablation on Re-Routing Methods** We further evaluate model performance under three re-routing strategies: to the most similar expert, to a random expert, and to the least similar expert. As shown in Table 7, re-routing to the most similar expert consistently outperforms random expert selection,

| Combine | Metric | K=1 | | | | K=2 | | | |
|---|---|---|---|---|---|---|---|---|---|
| | | Exam | Math | Code | AVG | Exam | Math | Code | AVG |
| Concat | Frob | 58.41 | 24.09 | 19.54 | 34.01 | 60.69 | 39.01 | 35.72 | 45.14 |
| Concat | Cosine | 58.74 | 25.38 | 13.70 | 32.61 | 60.76 | 39.88 | 34.32 | 44.99 |
| Concat | CKA-L | 58.24 | 30.20 | 18.54 | 35.66 | 60.76 | 40.06 | 33.82 | 44.88 |
| Concat | CKA-R | 58.34 | 30.53 | 19.77 | 36.21 | 60.67 | 39.55 | 32.60 | 44.27 |
| Concat | CKA-P | 58.73 | 30.52 | 20.26 | 36.50 | 60.80 | 39.60 | 30.37 | 43.59 |
| **Mean±*Std*** | / | **58.49**$_{\pm0.21}$ | **28.14**$_{\pm2.82}$ | **18.36**$_{\pm2.40}$ | **35.00**$_{\pm1.47}$ | **60.74**$_{\pm0.05}$ | **39.62**$_{\pm0.36}$ | **33.37**$_{\pm1.80}$ | **44.57**$_{\pm0.57}$ |
| Logic | Frob | 58.94 | 23.52 | 17.03 | 33.16 | 61.00 | 38.36 | 33.41 | 44.26 |
| Logic | Cosine | 58.89 | 29.19 | 17.74 | 35.27 | 60.61 | 40.43 | 31.69 | 44.24 |
| Logic | CKA-L | 58.02 | 28.91 | 18.55 | 35.16 | 60.72 | 39.52 | 31.77 | 44.00 |
| Logic | CKA-R | 57.76 | 28.55 | 17.13 | 34.48 | 60.57 | 38.24 | 35.34 | 44.72 |
| Logic | CKA-P | 58.42 | 28.49 | 16.92 | 34.61 | 60.89 | 40.29 | 32.60 | 44.59 |
| **Mean±*Std*** | / | **58.41**$_{\pm0.47}$ | **27.73**$_{\pm2.12}$ | **17.47**$_{\pm0.61}$ | **34.54**$_{\pm0.75}$ | **60.76**$_{\pm0.16}$ | **39.37**$_{\pm0.93}$ | **32.96**$_{\pm1.34}$ | **44.36**$_{\pm0.26}$ |

Table 5: Comparisons across different data-free similarity measures on Qwen1.5-MoE-A2.7B.

| Calibration Dataset | Volume | K=1 | | | | K=2 | | | |
|---|---|---|---|---|---|---|---|---|---|
| | | Exam | Math | Code | AVG | Exam | Math | Code | AVG |
| Fineweb | 400×128 | 57.55 | 31.02 | 18.64 | 37.87 | 60.48 | 40.87 | 36.58 | 47.15 |
| C4 | 400×128 | 57.42 | 30.82 | 17.54 | 37.85 | 60.87 | 41.21 | 35.24 | 47.09 |
| WIKI | 400×128 | 57.64 | 30.93 | 18.34 | 37.90 | 60.70 | 40.92 | 35.65 | 47.02 |
| **Mean±*Std*** | / | **57.54**$_{\pm0.11}$ | **30.92**$_{\pm0.10}$ | **18.17**$_{\pm0.57}$ | **37.87**$_{\pm0.03}$ | **60.68**$_{\pm0.20}$ | **41.00**$_{\pm0.18}$ | **35.82**$_{\pm0.69}$ | **47.09**$_{\pm0.07}$ |
| Exam | 400×128 | 58.20 | 33.34 | 22.88 | 38.14 | 60.80 | 40.26 | 35.34 | 45.47 |
| Math | 400×128 | 58.15 | 32.48 | 23.50 | 38.04 | 60.88 | 40.25 | 36.43 | 45.85 |
| Code | 400×128 | 58.58 | 33.06 | 23.70 | 38.45 | 60.71 | 40.63 | 36.75 | 46.03 |
| OpenCompass | 400×128 | 57.67 | 32.07 | 24.60 | 38.11 | 61.16 | 41.65 | 37.78 | 46.86 |
| **Mean±*Std*** | / | **58.15**$_{\pm0.38}$ | **32.74**$_{\pm0.50}$ | **23.67**$_{\pm0.63}$ | **38.19**$_{\pm0.15}$ | **60.89**$_{\pm0.18}$ | **40.70**$_{\pm0.61}$ | **36.58**$_{\pm1.03}$ | **46.05**$_{\pm0.61}$ |
| Fineweb | 200×64 | 57.54 | 30.93 | 17.54 | 37.56 | 60.82 | 40.20 | 35.12 | 46.66 |
| Fineweb | 400×128 | 57.55 | 31.02 | 18.64 | 37.87 | 60.48 | 40.87 | 36.58 | 47.15 |
| Fineweb | 800×256 | 57.95 | 31.88 | 17.53 | 38.06 | 60.44 | 40.87 | 34.34 | 46.58 |
| **Mean±*Std*** | / | **57.68**$_{\pm0.23}$ | **31.28**$_{\pm0.53}$ | **17.90**$_{\pm0.64}$ | **37.83**$_{\pm0.25}$ | **60.58**$_{\pm0.22}$ | **40.65**$_{\pm0.38}$ | **35.35**$_{\pm1.12}$ | **46.80**$_{\pm0.31}$ |

Table 6: Comparisons across different calibration datasets on Qwen1.5-MoE-A2.7B.

whereas choosing the least similar expert severely degrades performance. We also provide a theoretical analysis showing that similarity-based re-routing method yields a tighter upper bound on output perturbation (Appendix A.5). These results demonstrate the critical role of the similarity matrix in guiding effective expert selection.

| Method | K=1 | | | | K=2 | | | |
|---|---|---|---|---|---|---|---|---|
| | Exam | Math | Code | AVG | Exam | Math | Code | AVG |
| Most Sim | 57.55 | 31.02 | 18.64 | 37.87 | 60.66 | 40.87 | 36.58 | 47.15 |
| Random | 45.18 | 21.41 | 11.09 | 27.74 | 57.12 | 34.91 | 28.66 | 41.68 |
| Dis Sim | 11.03 | 1.55 | 0.00 | 4.72 | 38.77 | 28.05 | 9.65 | 28.74 |

Table 7: Comparisons across different re-routing methods on Qwen1.5-MoE-A2.7B.

## 5 CONCLUSION

In this work, we investigate the challenges faced by MoE models during batched inference. We analyze the expert similarity patterns and activation weight distributions in MoE models. Building on the insights, we propose SERE, a novel method for accelerating batched decoding in MoE models. SERE dynamically re-routes tokens assigned to secondary experts toward their most similar primary experts, thereby reducing the number of active experts, while preserving critical experts to safeguard model capability. We further develop a customized, efficient CUDA kernel for SERE. Extensive experiments demonstrate that SERE achieves up to $2\times$ speedup with only a slight impact on model quality. Our study provides new insights into MoE inference optimization, highlighting re-routing as a promising direction beyond traditional approaches such as pruning or quantization, and sets the stage for future work on dynamic expert selection and efficient MoE deployment.

ACKNOWLEDGEMENT

This work was supported in part by the Natural Science Foundation of China (No. 62332002, 62425101), The Guangdong Grants (Grant No.2023ZT10X075), and Shenzhen Science and Technology Program (KQTD20240729102051063).

REPRODUCIBILITY STATEMENT

We have made significant efforts to ensure the reproducibility of our work. The full implementation of SERE, including the efficient CUDA kernel, is available in the supplementary material and will be released upon publication. All experimental details, including model configurations, hyperparameters, and evaluation benchmarks, are thoroughly documented in Section 4.1 and Appendix B. The expert similarity matrices were computed using standard metrics (Frobenius, Cosine, CKA), as described in Appendix A.2. The calibration datasets (FineWeb, C4, WIKI, OpenCompass) are publicly available. Pseudocode for both similarity estimation (Algorithm 1) and the re-routing mechanism (Algorithm 2) is provided to facilitate replication. Our results can be reproduced using the described setup, and all relevant code and scripts can be found in `https://github.com/JL-Cheng/SERE`.

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

# A APPENDIX ON METHOD

## A.1 PRELIMINARIES ON MOES

The MoE architecture enhances model capacity and computational efficiency by conditionally activating only a subset of parameters for each token (Shazeer et al., 2017). An MoE layer consists of a set of expert networks $\{\mathbf{E}_1, \mathbf{E}_2, \ldots, \mathbf{E}_M\}$ and a router network $\mathbf{R}$. Given an input token embedding $\mathbf{x}$, the router produces routing logits $\mathbf{R}(\mathbf{x})$, which are transformed into a probability distribution over experts. The output activation $\mathbf{y}$ of an MoE layer can be expressed as:

$$\mathbf{y} = \sum_{i=1}^{M} \mathbf{P}_i(\mathbf{x}) \cdot \mathbf{E}_i(\mathbf{x}), \tag{6}$$

$$\mathbf{E}(\mathbf{x}) = (\sigma(\mathbf{x}\mathbf{W}_{\text{gate}}) \odot (\mathbf{x}\mathbf{W}_{\text{up}})) \mathbf{W}_{\text{down}}, \tag{7}$$

where $\mathbf{W}_{\text{gate}}, \mathbf{W}_{\text{up}} \in \mathbb{R}^{d_h \times d_m}$, $\mathbf{W}_{\text{down}} \in \mathbb{R}^{d_m \times d_h}$, $\sigma(\cdot)$ denotes the activation function, $\odot$ denotes element-wise multiplication, and $P_i(x)$ denotes the normalized routing weight assigned to expert $\mathbf{E}_i$. In practice, a top-$k$ gating strategy is often adopted to reduce computation. Specifically, only the $k$ experts with the largest routing logits are selected:

$$\mathbf{P}_i(\mathbf{x}) = \text{Softmax}(\text{Top-}k(\mathbf{R}_i(\mathbf{x}))). \tag{8}$$

MoE architectures achieve efficient scaling while maintaining strong performance, making them a widely adopted paradigm in modern large-scale Transformer-based models (Bai et al., 2023; Liu et al., 2024b; Yang et al., 2025a; Agarwal et al., 2025). Our proposed SERE method builds on standard MoE architectures and changes only the decoding-time routing targets, preserving all parameters and layer structures.

## A.2 EXPERT SIMILARITY METRICS

### A.2.1 COSINE SIMILARITY

Given activation matrices $\mathbf{X}_{\mathbf{E}}, \mathbf{X}_{\mathbf{F}} \in \mathbb{R}^{n \times d}$ from two experts $\mathbf{E}$ and $\mathbf{F}$, the cosine similarity is computed by averaging the instance-wise cosine similarities between their outputs. For each input $i$, let $\mathbf{x}_{\mathbf{E}}^{(i)}$ and $\mathbf{x}_{\mathbf{F}}^{(i)}$ denote the $i$-th row vectors of $\mathbf{X}_{\mathbf{E}}$ and $\mathbf{X}_{\mathbf{F}}$. The overall cosine similarity is computed by

$$\mathcal{M}_{\cos}(\mathbf{E}, \mathbf{F}) = \frac{1}{n} \sum_{i=1}^{n} \frac{\langle \mathbf{x}_{\mathbf{E}}^{(i)}, \mathbf{x}_{\mathbf{F}}^{(i)} \rangle}{\|\mathbf{x}_{\mathbf{E}}^{(i)}\|_2 \|\mathbf{x}_{\mathbf{F}}^{(i)}\|_2}. \tag{9}$$

### A.2.2 FROBENIUS SIMILARITY

We measure Frobenius similarity between two experts $\mathbf{E}$ and $\mathbf{F}$ by first calculating the Frobenius norm of the difference between their activation matrices, and then normalizing this value by the maximum norm across all expert pairs. Let

$$x_{\mathbf{E},\mathbf{F}} = \|\mathbf{X}_{\mathbf{E}} - \mathbf{X}_{\mathbf{F}}\|_F, \tag{10}$$

and let $\max(x)$ denote the maximum $x_{\mathbf{E},\mathbf{F}}$ among all pairs $(\mathbf{E}, \mathbf{F})$. The normalized Frobenius similarity is then given by

$$\mathcal{M}_{\text{fro}}(\mathbf{E}, \mathbf{F}) = 1 - \frac{x_{\mathbf{E},\mathbf{F}}}{\max(x)}. \tag{11}$$

This formulation ensures that the most similar expert pair achieves a score close to 1, while the least similar pair approaches 0.

### A.2.3 CENTERED KERNEL ALIGNMENT

Centered Kernel Alignment (CKA) (Kornblith et al., 2019) is a widely used metric for quantifying the similarity between neural representations, as it is invariant to isotropic scaling and orthogonal transformations. CKA computes the similarity between two sets of expert representations by comparing their Gram matrices constructed with a chosen kernel function. In our experiments, we consider three types of kernels: linear, RBF (Gaussian), and polynomial.

Given $\mathbf{X_E}, \mathbf{X_F} \in \mathbb{R}^{n \times d}$, the CKA similarity is defined by

$$\mathcal{M}_{\mathrm{CKA}}(\mathbf{E}, \mathbf{F}) = \frac{\mathrm{HSIC}(\mathbf{K_E}, \mathbf{K_F})}{\sqrt{\mathrm{HSIC}(\mathbf{K_E}, \mathbf{K_E})\,\mathrm{HSIC}(\mathbf{K_F}, \mathbf{K_F})}}, \tag{12}$$

where $\mathbf{K_E}$ and $\mathbf{K_F}$ are $n \times n$ Gram matrices computed by kernel $k(\cdot, \cdot)$:

- **Linear Kernel:**

$$\mathbf{K_E} = \mathbf{X_E}\mathbf{X_E}^{\top}, \quad \mathbf{K_F} = \mathbf{X_F}\mathbf{X_F}^{\top}. \tag{13}$$

- **RBF (Gaussian) Kernel:**

$$[\mathbf{K_E}]_{ij} = \exp\left(-\frac{\|\mathbf{x}_{\mathbf{E}}^{(i)} - \mathbf{x}_{\mathbf{E}}^{(j)}\|_2^2}{2\sigma^2}\right), \quad [\mathbf{K_F}]_{ij} = \exp\left(-\frac{\|\mathbf{x}_{\mathbf{F}}^{(i)} - \mathbf{x}_{\mathbf{F}}^{(j)}\|_2^2}{2\sigma^2}\right), \tag{14}$$

where $\sigma$ is the bandwidth parameter.

- **Polynomial Kernel:**

$$[\mathbf{K_E}]_{ij} = \left(\mathbf{x}_{\mathbf{E}}^{(i)\top}\mathbf{x}_{\mathbf{E}}^{(j)} + c\right)^d, \quad [\mathbf{K_F}]_{ij} = \left(\mathbf{x}_{\mathbf{F}}^{(i)\top}\mathbf{x}_{\mathbf{F}}^{(j)} + c\right)^d, \tag{15}$$

where $c$ is a constant and $d$ is the degree of the polynomial.

Here, HSIC denotes the Hilbert-Schmidt Independence Criterion, which measures the dependence between two Gram matrices. For practical implementation, we use the unbiased HSIC estimator as introduced by Kim & Han (2023), which provides $O(n^2)$ computational complexity.

### A.3 PARAMETER-BASED SIMILARITY COMPUTATION METHODS

We implement some data-free, parameter-based methods for computing expert similarities to compare against the activation-based methods. Considering each expert consists of three weight matrices, namely $\theta_1 = \mathbf{W}_{\mathrm{up}}$, $\theta_2 = \mathbf{W}_{\mathrm{gate}}$, and $\theta_3 = \mathbf{W}_{\mathrm{down}}$, we follow Zhang et al. (2025b) and apply two parameter combination strategies to merge these weights:

- **Concat**: The three weight matrices are directly concatenated: $\{\theta_1, \theta_2, \theta_3\}$. This method treats all weights equally without considering their functional roles in expert computation.
- **Logic**: The three weight matrices are combined according to the computational structure of an MoE expert, expressed as $\theta_3(\theta_1 \cdot \theta_2)$. This approach reflects the structural dependency among the three components.

After obtaining the combined expert weights, we compute the similarity matrices using the similarity metrics described in Appendix A.2. As discussed in Section 4.4, the parameter-based methods perform noticeably worse than the activation-based methods. Although parameter-based approaches are data-free, they are less effective at capturing the functional redundancy among experts.

### A.4 PSEUDOCODE

We provide the pseudocode for expert similarity estimation (Algorithm 1) and the CUDA-accelerated implementation of SERE (Algorithm 2) to facilitate readers' understanding of our approach. The pseudocode presents the key computational steps, helping to bridge the gap between the conceptual description and its practical realization.

### A.5 THEORETICAL ANALYSIS

In this section, we provide the theoretical justification that similarity-based expert re-routing can better preserve model capabilities.

**Definition 1** (MoE Layer Structure). *Consider a MoE model composed of $k$ MoE layers, where the $i$-th layer consists of $M$ experts $\{\mathbf{E}_1^{(i)}, \mathbf{E}_2^{(i)}, \ldots, \mathbf{E}_M^{(i)}\}$. For an input $z \in \mathbb{R}^d$, the layer output is a convex combination:*

$$\mathcal{N}_i(z) = \sum_{m=1}^{M} w_m^{(i)}(z) \cdot \mathbf{E}_m^{(i)}(z), \tag{16}$$

where $w_m^{(i)}(z) \geq 0$ and $\sum_{m=1}^{M} w_m^{(i)}(z) = 1$ *are routing weights determined by a router function. Let* $\mathcal{D}_0$ *denote the input data distribution, and* $\mathcal{D}_i$ *be the induced distribution of inputs to layer* $i$, *obtained by propagating samples* $x \sim \mathcal{D}_0$ *through the preceding layers.*

**Definition 2** (Expert Similarity). *For two experts* $\mathbf{E}_a^{(i)}$ *and* $\widetilde{\mathbf{E}}_a^{(i)}$ *at position* $a$ *in layer* $i$, *their similarity under the input distribution* $\mathcal{D}_i$ *is defined as:*

$$\delta(\mathbf{E}_a^{(i)}, \widetilde{\mathbf{E}}_a^{(i)}) = \mathbb{E}_{z \sim \mathcal{D}_i} \left[ \|\mathbf{E}_a^{(i)}(z) - \widetilde{\mathbf{E}}_a^{(i)}(z)\|_2 \right]. \tag{17}$$

**Theorem 1** (Expert Substitution Error Bound). *We consider replacing a single expert* $\mathbf{E}_a^{(i)}$ *in layer* $i$ *with another expert* $\widetilde{\mathbf{E}}_a^{(i)}$ *while keeping all other experts and routing weights unchanged, yielding a modified layer* $\widetilde{\mathcal{N}}_i$. *Let* $\mathcal{F} = \mathcal{N}_k \circ \cdots \circ \mathcal{N}_1$ *be the original network, and* $\widetilde{\mathcal{F}} = \mathcal{N}_k \circ \cdots \circ \widetilde{\mathcal{N}}_i \circ \cdots \circ \mathcal{N}_1$ *be the network with expert* $\mathbf{E}_a^{(i)}$ *replaced by* $\widetilde{\mathbf{E}}_a^{(i)}$. *Assume each downstream module* $\mathcal{N}_j$ *for* $j = i+1, \ldots, k$ *is Lipschitz continuous with constant* $L_j$, *and define* $\Lambda = \prod_{j=i+1}^{k} L_j$. *Let* $w_a^{(i)}(z)$ *be the routing weight assigned to expert* $a$. *Then the substitution error satisfies*

$$E(\widetilde{\mathbf{E}}_a^{(i)}, i) \leq \Lambda \cdot \mathbb{E}_{z \sim \mathcal{D}_i} \left[ w_a^{(i)}(z) \cdot \|\mathbf{E}_a^{(i)}(z) - \widetilde{\mathbf{E}}_a^{(i)}(z)\|_2 \right] \leq \Lambda \cdot \delta(\mathbf{E}_a^{(i)}, \widetilde{\mathbf{E}}_a^{(i)}), \tag{18}$$

*where the substitution error is*

$$E(\widetilde{\mathbf{E}}_a^{(i)}, i) = \mathbb{E}_{x \sim \mathcal{D}_0} \left[ \|\mathcal{F}(x) - \widetilde{\mathcal{F}}(x)\|_2 \right]. \tag{19}$$

*Proof.* For any $x \sim \mathcal{D}_0$, let $z_i = (\mathcal{N}_{i-1} \circ \cdots \circ \mathcal{N}_1)(x) \sim \mathcal{D}_i$. The layer output difference is

$$\mathcal{N}_i(z_i) - \widetilde{\mathcal{N}}_i(z_i) = w_a^{(i)}(z_i) \left( \mathbf{E}_a^{(i)}(z_i) - \widetilde{\mathbf{E}}_a^{(i)}(z_i) \right). \tag{20}$$

Let $\mathcal{G} = \mathcal{N}_k \circ \cdots \circ \mathcal{N}_{i+1}$, which is $\Lambda$-Lipschitz. Then,

$$\begin{aligned}
\|\mathcal{F}(x) - \widetilde{\mathcal{F}}(x)\|_2 &= \left\| \mathcal{G}(\mathcal{N}_i(z_i)) - \mathcal{G}(\widetilde{\mathcal{N}}_i(z_i)) \right\|_2 \\
&\leq \Lambda \cdot \|\mathcal{N}_i(z_i) - \widetilde{\mathcal{N}}_i(z_i)\|_2 \\
&= \Lambda \cdot w_a^{(i)}(z_i) \cdot \|\mathbf{E}_a^{(i)}(z_i) - \widetilde{\mathbf{E}}_a^{(i)}(z_i)\|_2.
\end{aligned} \tag{21}$$

Taking expectation over $x \sim \mathcal{D}_0$ gives

$$\begin{aligned}
E(\widetilde{\mathbf{E}}_a^{(i)}, i) &\leq \Lambda \cdot \mathbb{E}_{z_i \sim \mathcal{D}_i} \left[ w_a^{(i)}(z_i) \cdot \|\mathbf{E}_a^{(i)}(z_i) - \widetilde{\mathbf{E}}_a^{(i)}(z_i)\|_2 \right] \\
&\leq \Lambda \cdot \mathbb{E}_{z_i \sim \mathcal{D}_i} \left[ \|\mathbf{E}_a^{(i)}(z_i) - \widetilde{\mathbf{E}}_a^{(i)}(z_i)\|_2 \right] \\
&= \Lambda \cdot \delta(\mathbf{E}_a^{(i)}, \widetilde{\mathbf{E}}_a^{(i)}),
\end{aligned} \tag{22}$$

where the second inequality follows from $0 \leq w_a^{(i)}(z_i) \leq 1$. This completes the proof. $\square$

This analysis shows that the error bound of expert substitution is jointly determined by the structural stability of downstream layers ($\Lambda$) and the similarity between experts ($\delta(\cdot, \cdot)$). Therefore, under a fixed model architecture, re-routing tokens to a more similar expert yields a tighter upper bound on output perturbation. The above analysis provides theoretical support for the SERE method.

---

**Algorithm 1** Expert Similarity Estimation

---

**Input:** Calibration dataset $\mathcal{D}_{\text{calib}}$;
Number of iterations $N$;
Mixture-of-Experts (MoE) model with $L$ layers, each containing $M$ experts $\mathbf{E}_1^{(l)}, \ldots, \mathbf{E}_M^{(l)}$;
Similarity function $\text{Sim}(\cdot, \cdot)$
**Output:** Layer-wise similarity matrices $\{\mathbf{S}^{(l)} \in \mathbb{R}^{M \times M}\}_{l=1}^L$

**for** $l \leftarrow 1$ **to** $L$ **do**
 $\mathbf{S}^{(l)} \leftarrow \mathbf{0}_{M \times M}$ ;         // Initialize similarity matrix for layer $l$
**end**
**for** $i \leftarrow 1$ **to** $N$ **do**
 $\mathcal{B} \leftarrow$ the $i$-th batch from $\mathcal{D}_{\text{calib}}$;      // Load calibration dataset
 $\mathbf{X}^{(0)} \leftarrow \mathcal{B}$;           // Input to the first layer
 **for** $l \leftarrow 1$ **to** $L$ **do**
  **for** $j \leftarrow 1$ **to** $M$ **do**
   $\mathbf{A}_j^{(l)} \leftarrow \mathbf{E}_j^{(l)}\big(\mathbf{X}^{(l-1)}\big)$;   // Calculate activation for all experts.
  **end**
  **for** $p \leftarrow 1$ **to** $M$ **do**
   **for** $q \leftarrow p$ **to** $M$ **do**
    $s \leftarrow \text{Sim}\big(\mathbf{A}_p^{(l)}, \mathbf{A}_q^{(l)}\big)$;  // Accumulate pairwise similarities
    $\mathbf{S}^{(l)}[p, q] \mathrel{+}= s$;
    $\mathbf{S}^{(l)}[q, p] \mathrel{+}= s$;     // Ensure symmetry
   **end**
  **end**
  $\mathbf{X}^{(l)} \leftarrow \text{MoE}^{(l)}\big(\mathbf{X}^{(l-1)}\big)$;  // Standard MoE forward to get next layer input
 **end**
**end**
**for** $l \leftarrow 1$ **to** $L$ **do**
 $\mathbf{S}^{(l)} \leftarrow \mathbf{S}^{(l)}/N$ ;       // Normalize by number of iterations
**end**
**return** $\{\mathbf{S}^{(l)}\}_{l=1}^L$

---

# B APPENDIX ON EXPERIMENT SETTINGS

## B.1 MODELS

We evaluate SERE on three representative MoE models: Qwen1.5-MoE-A2.7B-Chat (Bai et al., 2023), DeepSeekV2-Lite (Liu et al., 2024b), and Qwen3-30B-A3B (Yang et al., 2025a).

**Qwen1.5-MoE-A2.7B-Chat:** Each token activates 4 shared experts and 4 routed experts (out of 60) in each layer.

**DeepSeekV2-Lite:** Each token activates 2 shared experts and 6 routed experts (out of 64) in each layer.

**Qwen3-30B-A3B:** Each token activates 8 routed experts (out of 128) in each layer.

More details can be found in Table 8.

## B.2 HYPER-PARAMETERS

For expert skipping, we evaluate two configurations that retain the Top-1 and Top-2 experts as the primary experts. For expert merging, we select pruning rates that yield TPOT comparable to that of expert skipping methods, ensuring a fair comparison. For SERE, similarity matrices are computed using the Frobenius norm on a calibration subset of FineWeb-Edu (Lozhkov et al., 2024) (400 se-

---

**Algorithm 2** CUDA-Accelerated SERE

---

**Input:** Top-K expert weights $\mathbf{W}^{(l)} \in \mathbb{R}^{T \times K}$;
Top-K expert indices $\mathbf{I}^{(l)} \in \mathbb{Z}^{T \times K}$;
Expert similarity matrix $\mathbf{S}^{(l)} \in \mathbb{R}^{M \times M}$;
Retain count $S \in [1, K)$;
Similarity threshold $\rho \in [0, 1]$
**Output:** Re-routed expert indices $\mathbf{I}'^{(l)} \in \mathbb{Z}^{T \times K}$
$\mathbf{I}'^{(l)} \leftarrow \mathbf{I}^{(l)}; \mathcal{H} \leftarrow \mathbf{0}_M;$           // Initialization
**for** $t \leftarrow 1$ **to** $T$ **and** $s \leftarrow 1$ **to** $S$ **do**
   $\mathcal{H}[I_{t,s}^{(l)}] \leftarrow 1;$           // Mark current (primary) expert as retained
**end**
$R_{total} \leftarrow T \times (K - S);$           // All secondary experts to be re-routed.
**for** *each CUDA thread* $tid \in [0, R_{total})$ *in parallel* **do**
   $t \leftarrow \lfloor tid/(K - S) \rfloor;$
   $k \leftarrow S + (tid \bmod (K - S));$           // Current token index
   **if** $t \geq T$ **or** $k \geq K$ **then return**;
   $e_{orig} \leftarrow \mathbf{I}_{t,k}^{(l)};$           // Original expert
   **if** $\mathcal{H}[e_{orig}] = 1$ **then**
      $\mathbf{I}_{t,k}'^{(l)} \leftarrow e_{orig};$           // No change if already retained
      **continue**;
   **end**
   $s_{best} \leftarrow -\infty, e_{best} \leftarrow 0;$           // Init maximum similarity and best matched expert
   **for** $e \leftarrow 0$ **to** $M - 1$ **do**
      **if** $\mathcal{H}[e] = 1$ **then**
         $s_{curr} \leftarrow S^{(l)}[e_{orig}, e];$           // Pairwise similarity with retained experts
         **if** $s_{curr} > s_{best}$ **then**
            $s_{best} \leftarrow s_{curr}, e_{best} \leftarrow e;$           // Update best similarity
         **end**
      **end**
   **end**
   **if** $\rho > 0$ **and** $s_{best} < \rho$ **then**
      $\mathbf{I}_{t,k}'^{(l)} \leftarrow e_{orig};$           // Keep original if below threshold
   **else**
      $\mathbf{I}_{t,k}'^{(l)} \leftarrow e_{best};$           // Re-route to the best matched retained expert
   **end**
**end**
**return** $\mathbf{I}'^{(l)}$

---

quences $\times$ 128 tokens). The similarity matrices are normalized to $[0, 1]$, where larger values indicate higher similarity between experts.

Tables 8 summarize the main inference configurations for all MoE models studied in this work. For the SERE method, the parameters `select_top_k` and `threshold` are tuned according to ablation and experimental requirements. All calibration and experiments are performed on NVIDIA H20 GPUs

## B.3 BENCHMARKS

For accuracy comparison, we select a diverse set of complex reasoning tasks from the OpenCompass benchmark (Contributors, 2023), covering multiple domains: **Exam** (CMMLU (Li et al., 2024), BoolQ (Clark et al., 2019), and BBH (Suzgun et al., 2023)); **Math** (Math (Hendrycks et al., 2021), GSM8K (Cobbe et al., 2021), and Math_401 (Yuan et al., 2023)); and **Code** (HumanEval (Chen et al., 2021), MBPP (Austin et al., 2021)). Because CMMLU and BoolQ are multiple-choice tasks, we adopt the CoT mode to evaluate the models' decoding capabilities. Details and examples of these tasks are provided in Table 9.

| Model Config | Qwen1.5-A2.7B-Chat | DeepSeekV2-Lite | Qwen3-30B-A3B |
|---|---|---|---|
| Total Params (B) | 14.3 | 16 | 30 |
| Activated Params (B) | 2.7 | 2.4 | 3 |
| MoE Layers / Total Layers | 24/24 | 26/27 | 48/48 |
| Experts per MoE Layer | 60 | 64 | 128 |
| Activated Experts per Token | 4 (selected) + 4 (shared) | 6 (selected) + 2 (shared) | 8 |
| hidden size | 2560 | 2048 | 2048 |
| intermediate size | 5632 | 10944 | 6144 |
| Vocabulary Size | 151936 | 102400 | 151936 |

| Inference Setting | Qwen1.5-A2.7B-Chat | DeepSeekV2-Lite | Qwen3-30B-A3B |
|---|---|---|---|
| Temperature | 0.7 | 0.3 | 0.7 |
| Top-$p$ | 0.8 | 0.95 | 0.8 |
| Top-$k$ | 20 | 50 | 20 |
| Repetition Penalty | 1.05 | 1.00 | 1.00 |
| Max Output Tokens | 1024 | 1024 | 2048 |
| Batch Size | 16 | 16 | 16 |

Table 8: Main inference hyperparameters for each model.

For acceleration comparison, we measure the online inference speed of different models under various methods using vLLM (Kwon et al., 2023). Each model is deployed on a single GPU, and we record the *Time per Output Token* (TPOT, in ms) across different *Queries per Second* (QPS) settings to emulate real-world service scenarios. The input and output sequence lengths are fixed at 128 and 32 tokens, respectively, and each test processes a total of 5,000 requests.

### B.4 CALIBRATION DATASET

In this work, we employ several calibration datasets to estimate expert similarity within MoE models, including three general datasets: FineWeb-Edu (Lozhkov et al., 2024), WIKI (Merity et al., 2017), C4 (Raffel et al., 2020), and four Domain-Specific datasets: Math, Code, Exam, and Open-Compass . These calibration sets are used to perform forward passes through the model, collecting activation values for each expert at every layer. The resulting activations are then utilized to compute inter-expert similarity metrics, which guide subsequent rerouting strategies.

**FineWeb-Edu** (Lozhkov et al., 2024) is a large-scale, high-quality English web corpus designed for pre-training and evaluation of language models. It contains diverse and well-filtered content, making it a representative resource for general-purpose calibration.

**WIKI** (Merity et al., 2017) refers to the English Wikipedia dump, a widely adopted dataset in NLP research. Its encyclopedic coverage and high linguistic quality make it suitable for calibrating models on general knowledge and formal text.

**C4** (Colossal Clean Crawled Corpus) (Raffel et al., 2020) is a massive web-crawled dataset filtered for high-quality English text. It is commonly used in large-scale language model pre-training and serves as a robust calibration set for open-domain language understanding.

**Math** is a domain-specific dataset constructed from Math (Hendrycks et al., 2021), GSM8K (Cobbe et al., 2021), and Math401 (Yuan et al., 2023) within OpenCompass. We randomly sample prompts and answers from these benchmarks and shuffle them to form the calibration set.

**Code** is a domain-specific dataset constructed from HumanEval (Chen et al., 2021) and MBPP (Austin et al., 2021) within OpenCompass. We randomly sample prompts and answers from these benchmarks and shuffle them to form the calibration set.

**Exam** is a domain-specific dataset constructed from CMMLU (Li et al., 2024), BoolQ (Clark et al., 2019) and BBH (Suzgun et al., 2023) within OpenCompass. We randomly sample prompts and answers from these benchmarks and shuffle them to form the calibration set.

**OpenCompass** combines the three domain-specific calibration datasets above and generates the calibration data through uniform sampling.

| Task | Domain/Format | Description / Example |
|---|---|---|
| **CMMLU** (Li et al., 2024) | Exam / Multiple-Choice | A comprehensive Chinese multi-subject exam benchmark with 57 subjects. 

 *Example:* 关系数据库中数据的逻辑结构是（A）树结构（B）维度表（C）层次结构（D）形状结构 |
| **BoolQ** (Clark et al., 2019) | Exam / Multiple-Choice (Yes/No) | Reading comprehension questions with yes/no answers based on a passage. 

 *Example:* Property tax – Property tax or 'house tax' is a local tax ... Is house tax and property tax are same? |
| **BBH** (Suzgun et al., 2023) | Exam / Diverse Reasoning | Big-Bench Hard, a collection of challenging tasks covering logical, symbolic, and commonsense reasoning. 

 *Example:* Which sentence has the correct adjective order: \n(A) medium-size archaic prismlike purple American car\n(B) archaic purple prismlike American medium-size car |
| **Math** (Hendrycks et al., 2021) | Math / Open-Ended | A dataset of high school-level mathematical problems requiring step-by-step solutions. 

 *Example:* A positive multiple of 45 less than 1000 is randomly selected. What is the probability that it is a two-digit integer? Express your answer as a common fraction. |
| **GSM8K** (Cobbe et al., 2021) | Math / Open-Ended | Grade school math word problems with a focus on multi-step reasoning. 

 *Example:* Shiloh is 44 years old today. In 7 years, he will be three times as old as his nephew. How old is his nephew today? |
| **Math_401** (Yuan et al., 2023) | Math / Open-Ended | MATH 401 is a benchmark dataset specifically designed to evaluate the arithmetic capabilities of large language models through a variety of arithmetic expressions and detailed performance analysis. 

 *Example:* 7.3947**2.5384= |
| **HumanEval** (Chen et al., 2021) | Code / Code Generation | Python programming problems requiring function implementation based on a natural language description. 

 *Example:* Write a function that returns the sum of two numbers. |
| **MBPP** (Austin et al., 2021) | Code / Code Generation | Mostly Basic Python Problems: Short Python programming tasks with input-output examples. 

 *Example:* Write a function to check if a string is a palindrome. |

Table 9: Overview of OpenCompass tasks used for evaluation.

For each calibration dataset, we randomly sample $N$ sequences and select a fixed number of tokens (Length) from each sequence. FineWeb-Edu, WIKI, and C4 are used as general-purpose calibration sets to evaluate SERE's performance under broad, diverse language phenomena, while Math, Code, Exam, and OpenCompass serve as task-specific calibration sets, aimed at testing whether downstream-oriented calibration data can further enhance SERE's capabilities, as well as the generalization or stability across different domains.

## C  Appendix on Experiments

### C.1  Detailed Analysis on Similarity Threshold

To better understand the relationship between the similarity threshold $\rho$ and model performance, we conduct a fine-grained empirical study on Qwen3-30B-A3B under $K = 1$ setting. Table 10 summarizes the performance across a range of $\rho$ values. The experimental results show that reasoning intensive tasks, such as mathematical problem solving and code generation, require higher similarity threshold compared with knowledge oriented tasks such as exam. For example, when $\rho$ reaches $0.5$, the performance on Exam benchmarks is already close to the baseline, while the performance on Math and Code benchmarks still exhibits a noticeable gap. This suggests that complex reasoning relies more critically on high-fidelity expert routing than factual recall.

| Threshold | cmmlu | boolq | bbh | math | gsm8k | math401 | heval | mbpp | avg | TPOT |
|---|---|---|---|---|---|---|---|---|---|---|
| 0.0 | 60.53 | 85.08 | 57.64 | 46.98 | 52.08 | 52.12 | 32.32 | 31.40 | 52.27 | 28.04 |
| 0.1 | 60.79 | 85.20 | 56.55 | 46.54 | 51.10 | 54.11 | 34.15 | 34.20 | 52.83 | 29.19 |
| 0.2 | 62.83 | 85.90 | 58.46 | 47.28 | 52.08 | 51.62 | 35.37 | 32.60 | 53.27 | 30.72 |
| 0.3 | 65.24 | 85.60 | 59.17 | 47.90 | 53.37 | 54.11 | 39.63 | 32.40 | 54.68 | 29.21 |
| 0.4 | 72.11 | 87.61 | 61.78 | 48.56 | 53.30 | 54.61 | 45.12 | 34.20 | 57.16 | 29.81 |
| 0.5 | 77.89 | 89.76 | 65.45 | 53.40 | 54.28 | 54.86 | 64.02 | 53.20 | 64.11 | 33.10 |
| 0.6 | 80.77 | 89.91 | 71.33 | 59.56 | 63.00 | 63.34 | 83.54 | 68.80 | 72.53 | 34.28 |
| 0.7 | 80.92 | 90.31 | 74.87 | 70.24 | 88.40 | 82.04 | 86.59 | 74.20 | 80.95 | 35.37 |
| 0.8 | 84.08 | 89.94 | 76.10 | 70.92 | 89.39 | 81.30 | 86.59 | 76.60 | 81.86 | 38.92 |
| 0.9 | 84.33 | 89.82 | 76.66 | 72.42 | 89.61 | 79.05 | 86.59 | 75.60 | 81.76 | 46.02 |
| 1.0 | 84.92 | 89.82 | 76.62 | 72.46 | 88.93 | 81.30 | 88.41 | 78.00 | 82.56 | 44.54 |

Table 10: Performance of Qwen3-30B-A3B under $K = 1$ setting across different thresholds.

In summary, the similarity threshold serves as a principled mechanism to balance efficiency and model performance. The empirical results suggest that setting $\rho$ to moderate or high values significantly improves performance on challenging tasks, primarily by eliminating a part of detrimental set of low-similarity rerouting decisions.

### C.2  Detailed Analysis on Prefilling Stage

SERE is primarily designed to accelerate the batched decoding phase of MoE models. By reducing the number of activated experts, it lowers the memory-communication overhead and thus speeds up the memory-bound decoding process. Because it does not reduce the computation FLOPs, it is not expected to provide noticeable speedups in the compute-bound prefill stage. Nevertheless, to give a more comprehensive understanding of SERE, we additionally conduct experiments evaluating its impact on the prefill stage, including its effect on prefill latency and the quality of the KV cache.

We first evaluated the **Time To First Token (TTFT)** of three MoE models: Qwen1.5-A2.7B-Chat, Qwen3-30B-A3B, and DeepSeekV2-Lite, under different QPS settings. As shown in Table 11, SERE achieves slightly lower TTFT than the baseline, but the improvement is marginal. The results are consistent with our expectations and also indicate that our CUDA-based re-routing implementation is highly efficient, introducing no additional overhead even when processing a large number of tokens during the prefill stage. In a typical generation scenario (e.g., 128 input tokens followed by 256 output tokens), prefill accounts for less than $1\%$ of the total latency, and this proportion will be even smaller when the outputs become longer. Therefore, we consider acceleration during the decoding stage to be substantially more impactful than acceleration during prefill.

We further examined whether SERE affects the KV cache generated during the prefill stage, since this could influence the quality of subsequent decoding. We first analyzed the proportion of primary experts among all activated experts under some typical batch settings. As shown in Table 12, for MoE models with fewer experts, such as Qwen1.5-A2.7B-Chat and DeepSeekV2-Lite, all activated experts are primary experts ($100\%$). Even for Qwen3-30B-A3B that has a larger number of experts, more than $80\%$ of the activated experts are retained as primary experts. These results indicate that nearly all activated experts are preserved as primary experts during prefill. Besides, the small number of secondary experts that require re-routing can also find similar substitutes more easily because the pool of primary experts is large. As a result, the impact on KV cache quality is minimal.

| Model / QPS | 8 | 16 | 24 | 32 |
|---|---|---|---|---|
| Qwen1.5-A2.7B-Chat | 33.64 | 40.62 | 45.33 | 51.43 |
| SERE ($K = 2$) | 33.57 | 38.53 | 45.24 | 50.24 |
| SERE ($K = 1$) | 32.48 | 37.64 | 42.58 | 48.10 |
| Qwen3-30B-A3B | 66.72 | 81.08 | 96.02 | 114.03 |
| SERE ($K = 2$) | 65.09 | 78.52 | 92.69 | 104.36 |
| SERE ($K = 1$) | 64.94 | 78.62 | 92.25 | 107.44 |
| DeepSeekV2-Lite | 67.06 | 82.57 | 93.12 | 106.44 |
| SERE ($K = 2$) | 66.03 | 79.99 | 91.10 | 108.23 |
| SERE ($K = 1$) | 66.04 | 79.60 | 92.67 | 107.61 |

Table 11: TTFT(ms) under varying QPS settings.

| Model / Batch Config | $32 \times 128$ | $16 \times 64$ | $4 \times 256$ |
|---|---|---|---|
| Qwen1.5-A2.7B-Chat | 100% | 100% | 100% |
| Qwen3-30B-A3B | 94.53% | 86.71% | 81.65% |
| DeepSeekV2-Lite | 100% | 100% | 100% |

Table 12: Percentage of primary experts retained during prefill.

| Methods \Tasks | Exam | | | Math | | | Code | | Avg. |
|---|---|---|---|---|---|---|---|---|---|
| | cmmlu | boolq | bbh | math | gsm8k | $math_{401}$ | heval | mbpp | (Acc. ↑) |
| Qwen3-30B-A3B $_{top8}$ | 84.88 | 90.21 | 76.70 | 72.28 | 89.23 | 79.05 | 87.20 | 78.40 | 82.24 |
| Qwen3-30B-A3B $_{top2}$ | 10.01 | 60.52 | 10.48 | 3.38 | 6.97 | 16.96 | 3.66 | 2.40 | 14.30 |
| SERE $_{top2;\ \rho=0.0}$ | 81.24 | 89.79 | 71.33 | 70.22 | 82.41 | 80.80 | 82.93 | 63.80 | 77.82 |
| **SERE (decode-only)** $_{top2;\ \rho=0.0}$ | 80.31 | 89.42 | 71.81 | 69.60 | 82.41 | 80.80 | 84.15 | 63.60 | 77.33 |
| SERE $_{top2;\ \rho=0.5}$ | 81.51 | **90.37** | **74.15** | **72.06** | **85.97** | **81.55** | 85.37 | **72.00** | **80.37** |
| **SERE (decode-only)** $_{top2;\ \rho=0.5}$ | **81.65** | 90.12 | 73.50 | 71.22 | 84.38 | 81.05 | **87.20** | 70.20 | 79.78 |
| Qwen3-30B-A3B $_{top1}$ | 0.00 | 61.68 | 4.89 | 0.08 | 0.91 | 1.25 | 0.00 | 0.00 | 8.60 |
| SERE $_{top1;\ \rho=0.0}$ | 60.53 | 85.08 | 57.64 | 46.98 | 52.08 | 52.12 | 32.32 | 31.40 | 52.27 |
| **SERE (decode-only)** $_{top1;\ \rho=0.0}$ | 62.96 | 85.02 | 57.56 | 46.32 | 50.95 | 52.37 | 37.80 | 32.60 | 51.20 |
| SERE $_{top1;\ \rho=0.5}$ | 77.89 | 89.76 | 65.45 | **53.40** | **54.28** | **54.86** | 64.02 | **53.20** | **64.11** |
| **SERE (decode-only)** $_{top1;\ \rho=0.5}$ | **78.68** | **89.82** | **65.60** | 52.48 | 53.68 | 53.62 | **66.46** | 51.00 | 63.34 |

Table 13: SERE vs. decode-only variant on Qwen3-30B-A3B across OpenCompass benchmarks.

To directly understand how SERE affects the KV cache produced during the prefill stage, we implemented and evaluated a **decode-only** variant in which all activated experts are preserved during prefill and re-routing is applied only during decoding. We tested this setting on the Qwen3-30B-A3B model across OpenCompass benchmarks, and the results are shown in Table 13. Surprisingly, across different skipping rates and thresholds, the decode-only variant consistently underperforms the original SERE method. We consider this may be because inconsistent expert selection between prefill and decoding stages introduces a distribution shift that particularly affects reasoning tasks that rely on stable internal representations.

In summary, although SERE does not provide significant acceleration during the prefill stage, it can be applied safely without degrading KV cache quality or overall performance.

## C.3 SIMILARITY MATRICES VISUALIZATION

In this section, we present a detailed visualization of the expert similarity matrices for Qwen1.5-2.7B (Bai et al., 2023), DeepSeekV2-Lite (Liu et al., 2024a), Qwen3-30B-A3B (Yang et al., 2025a), DeepSeekMoE (Dai et al., 2024), Ling-mini-2.0 (Li et al., 2025), and OLMoE-1B-7B-0125-Instruct (Muennighoff et al., 2025), as shown in Figure 9 to Figure 14, respectively. These visualizations reveal that different MoE architectures exhibit distinct similarity patterns across layers, i.e., some layers display highly clustered experts with strong intra-group similarity, whereas others show more uniform or dispersed similarity distributions. Such layer-specific variation indicates that the functional roles and redundancy levels of experts vary not only between models

but also across different layers within the same model, highlighting the importance of layer-wise analysis when designing expert routing or pruning strategies.

# D  LLM USAGE STATEMENT

In preparing this manuscript, we used LLMs solely to aid in polishing the writing, such as improving grammar, clarity, and readability. All substantive contributions to the research, including the conception of ideas, experimental design, data analysis, and so on, were made exclusively by the authors. The authors have thoroughly reviewed and taken responsibility for all content in the paper.

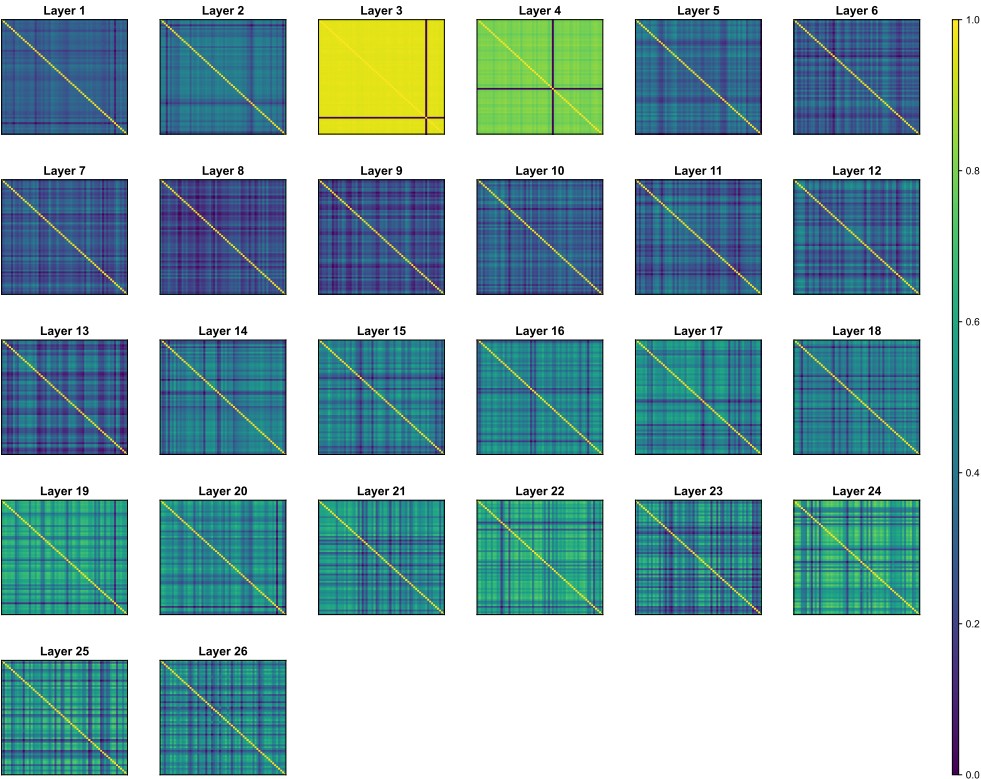

Figure 9: Visualization of expert similarity matrices of DeepSeekV2-Lite model.

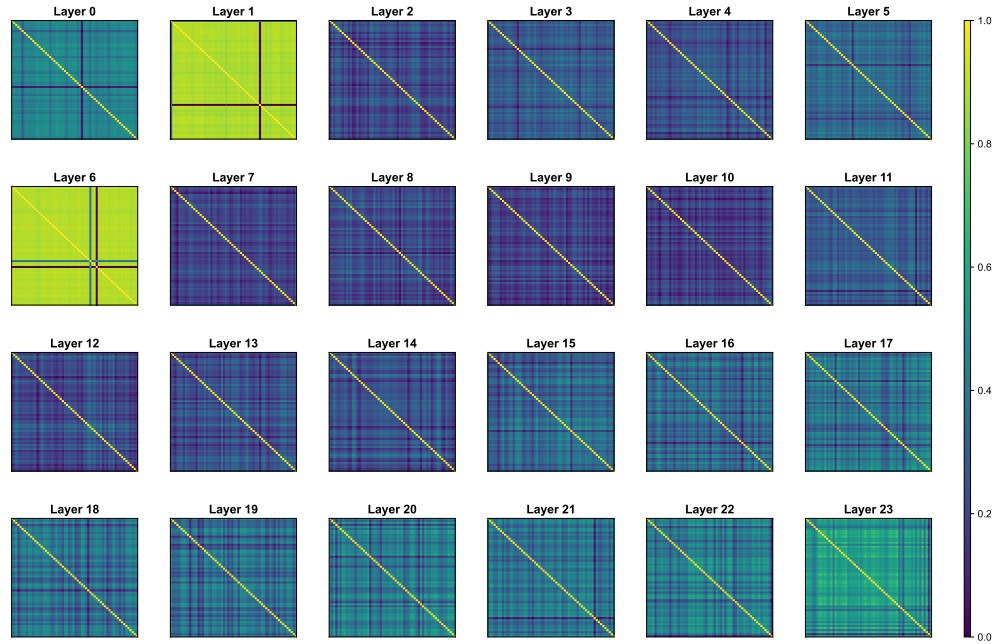

Figure 10: Visualization of expert similarity matrices of Qwen1.5-A2.7B model.

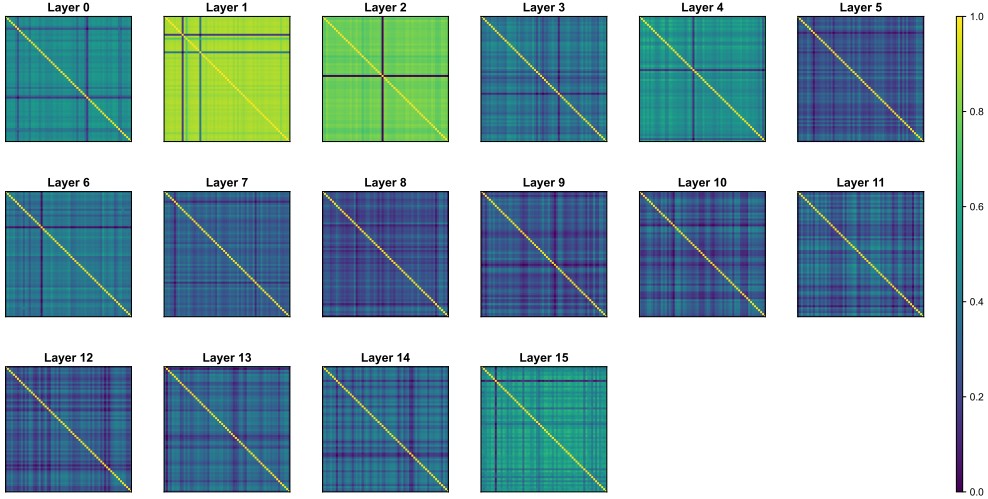

Figure 11: Visualization of expert similarity matrices of OLMoE-1B-7B-0125-Instruct model.

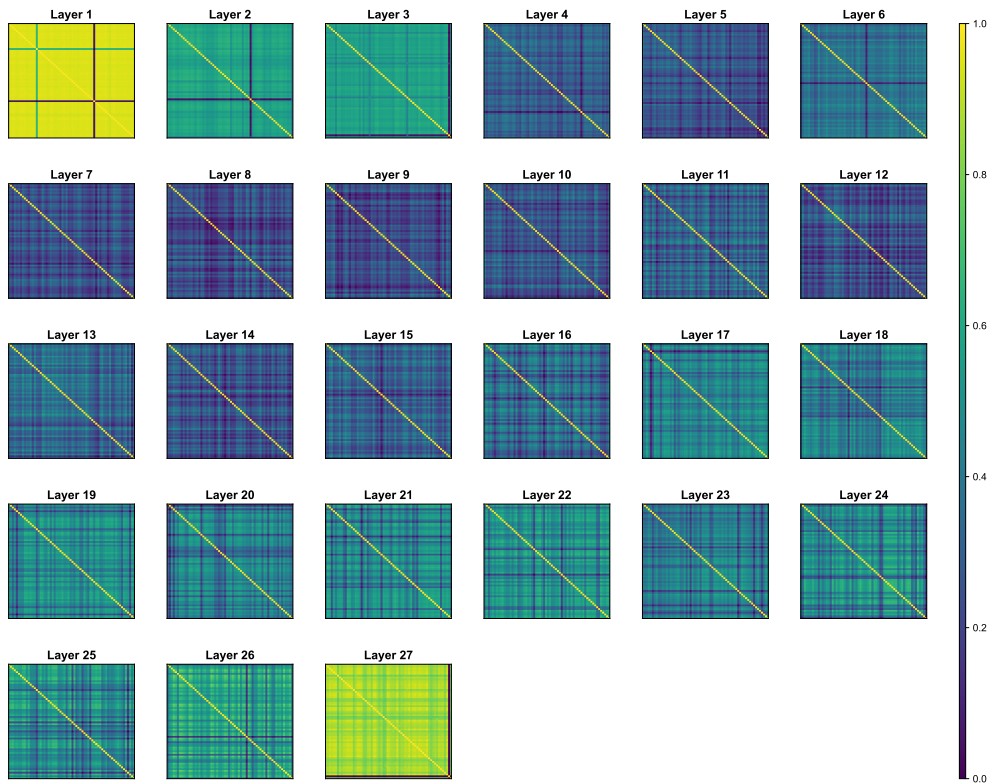

Figure 12: Visualization of expert similarity matrices of DeepSeekMoE model.

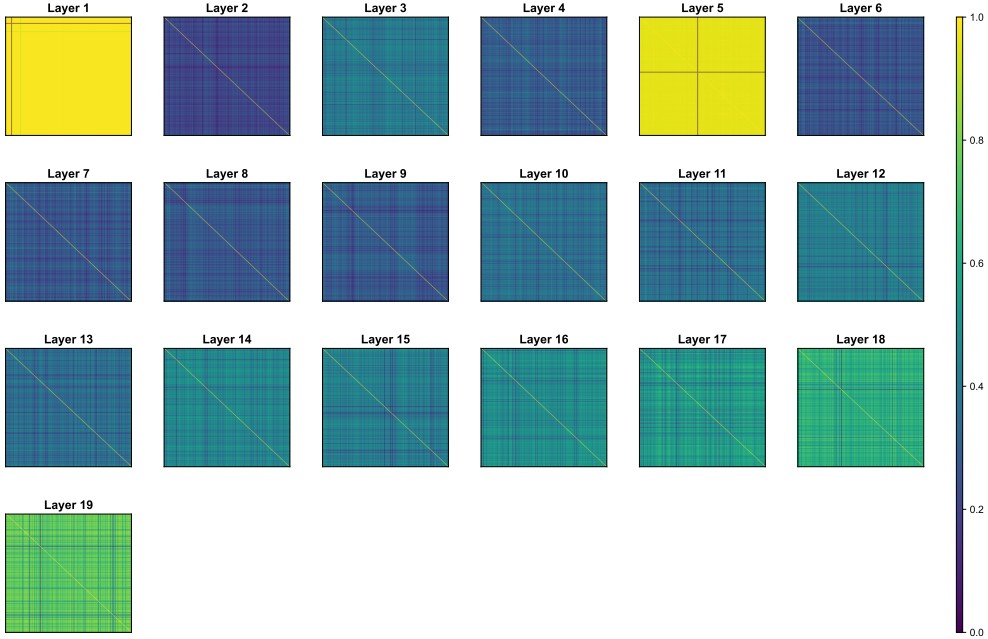

Figure 13: Visualization of expert similarity matrices of Ling-mini-2.0 model.

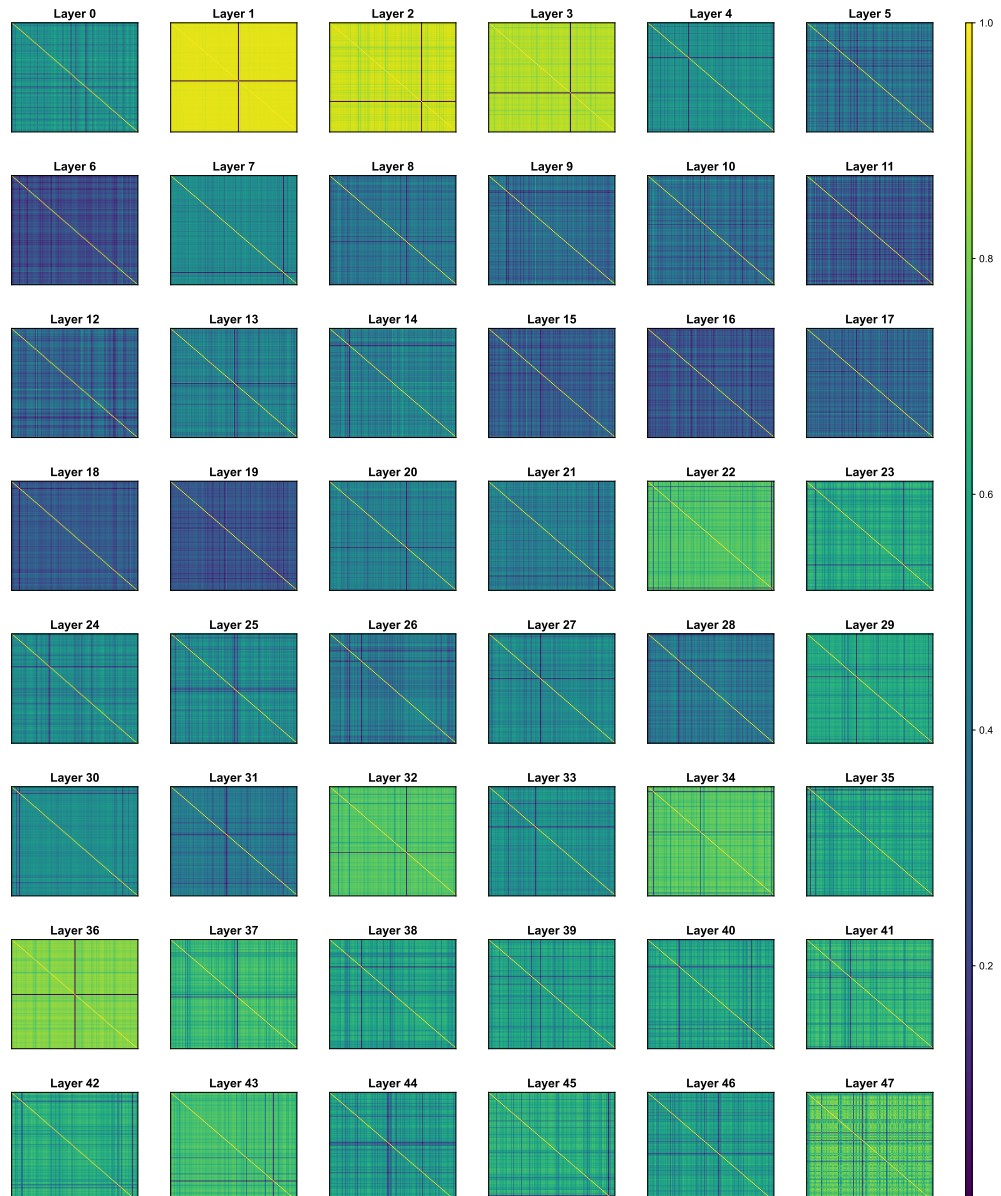

Figure 14: Visualization of expert similarity matrices of Qwen3-30B-A3B model.

