# OpenReview forum: "SERE: Similarity-based Expert Re-routing for Efficient Batch Decoding in MoE Models"
_ICLR.cc/2026/Conference — ICLR 2026 Poster_

### Official Review · Reviewer_2FF4 · 2025-10-25

**Soundness:** 4
**Presentation:** 3
**Contribution:** 3
**Rating:** 8
**Confidence:** 4

**Summary:**

The paper proposes SERE (Similarity-based Expert Re-Routing), a method to accelerate batch decoding in Mixture-of-Experts (MoE) large language models. The key idea is to identify and exploit functional similarity among experts to reduce redundant expert activation during inference. SERE precomputes an expert similarity matrix using a calibration dataset, then dynamically re-routes tokens from low-contribution (“secondary”) experts to their most similar “primary” experts while preserving dissimilar, critical ones. This reduces the number of active experts per batch without retraining or architectural changes.

The authors implement an efficient custom CUDA kernel compatible with vLLM, allowing seamless deployment with minimal code modification. Extensive experiments on several state-of-the-art MoE models (Qwen, DeepSeek) and reasoning, math, and coding benchmarks show up to 2× speedup in decoding with negligible accuracy loss.

**Strengths:**

The analysis in this paper is well-reasoned. It begins with a similarity analysis that reveals important insights into the relationships among MoE experts, which then guide subsequent research and design decisions. I like this style.


The kernel-level optimization and re-routing mechanism are well-aligned with system-level efficiency goals. This is an impressive aspect of the study.

Extensive experiments on multiple MoE architectures (Qwen, DeepSeek) and benchmarks (OpenCompass, MATH, HumanEval) show up to 2× decoding speedup with minimal accuracy loss (<3%).

**Weaknesses:**

The work is mostly empirical and engineering-oriented; it lacks theoretical analysis of why similarity-based re-routing preserves model behavior or formal guarantees on capacity preservation.

No analytical insight is given into how similarity thresholds interact with model generalization or stability.

**Questions:**

The similarity matrix is computed using a fixed calibration dataset (e.g., FineWeb-Edu).

How sensitive is SERE to domain shift or unseen data distributions? Would reusing the same similarity matrix across different domains (e.g., code vs. math) still maintain accuracy?

As I understand it, SERE first constructs a similarity matrix and then applies re-routing for the secondary experts. This is an interesting and promising idea. However, it would be valuable for the authors to discuss whether the dataset used to build the similarity matrix is representative enough and whether its distribution supports generalization. I do not consider this a weakness, but rather an opportunity for clarification and discussion.

---

> ### Author Response · Authors · 2025-11-20
> **Response to Reviewer 2FF4 [Part 1/2]**
>
> Dear Reviewer,  We appreciate your thorough evaluation of our paper. Below, we address your concerns and provide our responses. All revisions will be included in the forthcoming rebuttal version of the paper.
>
> ------
>
> **Q1: The work is mostly empirical and engineering-oriented; it lacks theoretical analysis of why similarity-based re-routing preserves model behavior or formal guarantees on capacity preservation.**
>
> **A1:** Thank you very much for your constructive suggestion. In response, we have added a theoretical analysis to the manuscript demonstrating that similarity-based expert re-routing effectively preserves model capacity. Specifically, under the assumption of Lipschitz continuity, we derive an upper bound on the output perturbation that depends on both the network’s stability and the similarity among experts. The detailed analysis can be found in "Appendix A.4: THEORETICAL ANALYSIS" in the revised manuscript.
>
> ------
>
> **Q2: No analytical insight is given into how similarity thresholds interact with model generalization or stability.**
>
> **A2:** We sincerely appreciate your insightful comment. In the SERE method, the similarity threshold plays a crucial role in balancing the number of “critical experts”. As discussed in Section 4.4 "ABLATION STUDY", we systematically investigate the impact of similarity threshold across different models. Our results demonstrate that a higher similarity threshold retains more experts, thereby enhancing model generalization and stability at the cost of reduced acceleration. We identify a threshold "sweet spot" that achieves an optimal trade-off between model speed and performance. Moreover, the experiments reveal that the influence of the similarity threshold varies noticeably across different models. However, even when no critical experts are preserved (i.e., $\rho = 0$), SERE still delivers strong performance across various MoE models and OpenCompass tasks, underscoring its inherent robustness and minimal negative impact on model generalization and stability.
>
> To further analyze how the similarity threshold affects generalization, we evaluate Qwen3-30B-A3B under K=1 setting across a range of thresholds on multiple tasks, with results summarized below:
>
> | **Threshold** | **cmmlu** | **boolq** | **bbh** | **math** | **gsm8k** | **math401** | **heval** | **mbpp** | **avg** |
> | ------------- | --------- | --------- | ------- | -------- | --------- | ----------- | --------- | -------- | ------- |
> | 0.0           | 60.53     | 85.08     | 57.64   | 46.98    | 52.08     | 52.12       | 32.32     | 31.40    | 52.27   |
> | 0.1           | 60.79     | 85.20     | 56.55   | 46.54    | 51.10     | 54.11       | 34.15     | 34.20    | 52.83   |
> | 0.2           | 62.83     | 85.90     | 58.46   | 47.28    | 52.08     | 51.62       | 35.37     | 32.60    | 53.27   |
> | 0.3           | 65.24     | 85.60     | 59.17   | 47.90    | 53.37     | 54.11       | 39.63     | 32.40    | 54.68   |
> | 0.4           | 72.11     | 87.61     | 61.78   | 48.56    | 53.30     | 54.61       | 45.12     | 34.20    | 57.16   |
> | 0.5           | 77.89     | 89.76     | 65.45   | 53.40    | 54.28     | 54.86       | 64.02     | 53.20    | 64.11   |
> | 0.6           | 80.77     | 89.91     | 71.33   | 59.56    | 63.00     | 63.34       | 83.54     | 68.80    | 72.53   |
> | 0.7           | 80.92     | 90.31     | 74.87   | 70.24    | 88.40     | 82.04       | 86.59     | 74.20    | 80.95   |
> | 0.8           | 84.08     | 89.94     | 76.10   | 70.92    | 89.39     | 81.30       | 86.59     | 76.60    | 81.86   |
> | 0.9           | 84.33     | 89.82     | 76.66   | 72.42    | 89.61     | 79.05       | 86.59     | 75.60    | 81.76   |
> | 1.0           | 84.92     | 89.82     | 76.62   | 72.46    | 88.93     | 81.30       | 88.41     | 78.00    | 82.56   |
>
> The results show that reasoning intensive tasks, such as mathematical problem solving and code generation, require higher similarity threshold compared with knowledge oriented tasks such as exam. For example, when $\rho$ reaches 0.5, the performance on Exam benchmarks is already close to the baseline, while the performance on Math and Code benchmarks still exhibits a noticeable gap. This suggests that complex reasoning relies more critically on high-fidelity expert routing than factual recall.
>
> These analysis have been incorporated into the revised manuscript (see Appendix C.1 "DETAILED ANALYSIS ON SIMILARITY THRESHOLD").

---

> ### Author Response · Authors · 2025-11-20
> **Response to Reviewer 2FF4 [Part 2/2]**
>
> **Q3: The similarity matrix is computed using a fixed calibration dataset (e.g., FineWeb-Edu). How sensitive is SERE to domain shift or unseen data distributions? Would reusing the same similarity matrix across different domains (e.g., code vs. math) still maintain accuracy?**
>
> **A3:** Thank you for raising this important and constructive question. We have conducted ablation studies to examine the sensitivity of SERE to the choice of calibration datasets. Specifically, we experimented on some general datasets (FineWeb-Edu, C4, WIKI) and domain-specific datasets (Exam, Math, Code, and the three mixed domains). Details of these datasets are provided in Appendix B.4. The general datasets are used to evaluate SERE's performance under broad, diverse language phenomena, while the domain-specific datasets aim at testing whether downstream‑oriented calibration data can further enhance SERE's capabilities, as well as the generalization or stability across different domains. The results are shown below:
>
> | **Calibration Set**      | **K=1**          |                  |                  |                  | **K=2**          |                  |                  |                  |
> | ------------------------ | ---------------- | ---------------- | ---------------- | ---------------- | ---------------- | ---------------- | ---------------- | ---------------- |
> |                          | **Exam**         | **Math**         | **Code**         | **AVG**          | **Exam**         | **Math**         | **Code**         | **AVG**          |
> | **General Data**         |                  |                  |                  |                  |                  |                  |                  |                  |
> | FineWeb                  | 57.55            | 31.02            | 18.64            | 37.87            | 60.48            | 40.87            | 36.58            | 47.15            |
> | C4                       | 57.42            | 30.82            | 17.54            | 37.85            | 60.87            | 41.21            | 35.24            | 47.09            |
> | WIKI                     | 57.64            | 30.93            | 18.34            | 37.90            | 60.70            | 40.92            | 35.65            | 47.02            |
> | **Mean ± Std**           | **57.54 ± 0.11** | **30.92 ± 0.10** | **18.17 ± 0.57** | **37.87 ± 0.03** | **60.68 ± 0.20** | **41.00 ± 0.18** | **35.82 ± 0.69** | **47.09 ± 0.07** |
> | **Domain-Specific Data** |                  |                  |                  |                  |                  |                  |                  |                  |
> | Exam                     | 58.20            | 33.34            | 22.88            | 38.14            | 60.80            | 40.26            | 35.34            | 45.47            |
> | Math                     | 58.15            | 32.48            | 23.50            | 38.04            | 60.88            | 40.25            | 36.43            | 45.85            |
> | Code                     | 58.58            | 33.06            | 23.70            | 38.45            | 60.71            | 40.63            | 36.75            | 46.03            |
> | OpenCompass              | 57.67            | 32.07            | 24.60            | 38.11            | 61.16            | 41.65            | 37.78            | 46.86            |
> | **Mean ± Std**           | **58.15 ± 0.38** | **32.74 ± 0.50** | **23.67 ± 0.63** | **38.19 ± 0.15** | **60.89 ± 0.18** | **40.70 ± 0.61** | **36.58 ± 1.03** | **46.05 ± 0.61** |
>
> As can be seen from the results, when K=2, the performance remains highly consistent across different calibration datasets, indicating that **SERE is robust with respect to the type of calibration data** . We also find that even when calibrated with domain-specific data, SERE maintains strong performance on other domains, which suggests that the similarity matrix captures transferable expert relationships rather than overfitting to a particular domain. Furthermore, when K=1, domain-specific calibration provides slightly better results than general calibration, indicating that in high skipping rate settings, using domain-specific calibration data can further improve the performance of SERE.
>
> These findings have been incorporated into the revised manuscript (see Section 4.4 "ABLATION STUDY" for details).

---

### Official Review · Reviewer_FvHj · 2025-10-31

**Soundness:** 3
**Presentation:** 3
**Contribution:** 3
**Rating:** 6
**Confidence:** 4

**Summary:**

The paper introduces SERE (Similarity-based Expert Re-routing), an inference-time method to accelerate batched decoding in MoE models. The key motivation is that the many experts activated across a batch in the decoding stage create a memory-bandwidth bottleneck. SERE addresses this by observing that router scores are often concentrated on a few "primary" experts. It proposes to re-route tokens assigned to lower-ranked "secondary" experts to their most functionally similar primary counterparts. This similarity is pre-computed offline using a data-driven approach on a calibration set. The method is designed to preserve "critical" experts that are dissimilar to others, preventing catastrophic capability loss. The authors demonstrate SERE's effectiveness on several modern MoE models, showing up to a 2x speedup in decoding with minimal accuracy degradation on reasoning benchmarks. One major contribution is the release of an efficient, plug-and-play CUDA kernel compatible with the vLLM inference server.

**Strengths:**

- The authors address a relevant problem that limits the efficiency and scalability of MoEs.
- The idea of the method is simple yet meaningful. The results indicate speedups up to 2x with acceptable accuracy drop.
- The authors provide an efficient kernel pluggable into vLLM.

**Weaknesses:**

**Reliance on and Sensitivity to Calibration Data:** The method's foundation is a similarity matrix computed on a calibration dataset. This introduces an offline computation step and a data dependency. While the ablations in Tables 4 and 5 show robustness to the choice of a general-domain dataset, the results also hint at a potential domain mismatch problem. For instance, the average similarity scores for the 'Code' domain are consistently lower than for others, suggesting that re-routing decisions might be less effective or even harmful for specialized domains not well-represented in the calibration data. The authors should (1) discuss this limitation and the potential need for domain-specific calibration, and (2) compare their data-driven similarity approach to a static, data-free alternative (e.g., using the cosine similarity of expert FFN weights) as a baseline. This would clarify the benefit of using activation-based similarity.

**Focus on Decoding:** The paper's focus and evaluation are exclusively on the batched decoding stage. The authors do not discuss or evaluate SERE's impact on the initial prefill stage, which is typically compute-bound. It is not clear if the method would provide any speedup in this stage, and it could potentially degrade the quality of the initial KV-cache generation. The authors need to discuss this in the paper.

**Questions:**

- In some MoE models, including Qwen3, there is a top-k weight normalization to adjust the total expert weights to 1. Do you keep this operation?
- Would this method work for the prefilling stage?
- It would be helpful to include standard deviation values to Tables 4 and 5 (AVG column).

---

> ### Author Response · Authors · 2025-11-20
> **Response to Reviewer FvHj [Part 1/4]**
>
> Thank you for your insightful and constructive feedback, which has significantly contributed to the improvement of our paper. Below, we address your concerns in detail. All the revisions will be included in the forthcoming rebuttal version of the paper.
>
> ------
>
> **Q1: Reliance on and Sensitivity to Calibration Data**
>
> **A1 [Part 1/2]:**  We sincerely appreciate your helpful comment. First, we clarify that the values reported under the “Code” column in Tables 4 and 5 correspond to the average task scores achieved by the SERE method, while not similarity scores. Specifically, for each row in Tables 4 and 5, we compute a single similarity matrix based on the specified method/calibration dataset and apply it during re-routing across all evaluation tasks. However, we fully agree that your points regarding **(1) domain-specific calibration** and **(2) data-free alternatives** are critical and require deeper discussion.
>
> **(1) Limitation and Potential Need for Domain-Specific Calibration**   In the original manuscript, we have already experimented on the OpenCompass calibration set. This calibration set is generated by randomly sampling from downstream task data, with the goal of providing a task‑oriented calibration set that better reflects the distribution of the target evaluation data. As shown in Table 5 in the original manuscript, although this calibration set performs slightly better than Fineweb-Edu at K=1, their performance becomes comparable at K=2.
>
> To further investigate domain-specific calibration, we conducted additional experiments using calibration data drawn exclusively from each target domain (Exam, Math, or Code). The results are summarized below:
>
> | Calibration Set          | K=1              |                  |                  |                  | K=2              |                  |                  |                  |
> | ------------------------ | ---------------- | ---------------- | ---------------- | ---------------- | ---------------- | ---------------- | ---------------- | ---------------- |
> |                          | **Exam**         | **Math**         | **Code**         | **AVG**          | **Exam**         | **Math**         | **Code**         | **AVG**          |
> | **General Data**         |                  |                  |                  |                  |                  |                  |                  |                  |
> | FineWeb                  | 57.55            | 31.02            | 18.64            | 37.87            | 60.48            | 40.87            | 36.58            | 47.15            |
> | C4                       | 57.42            | 30.82            | 17.54            | 37.85            | 60.87            | 41.21            | 35.24            | 47.09            |
> | Wiki                     | 57.64            | 30.93            | 18.34            | 37.90            | 60.70            | 40.92            | 35.65            | 47.02            |
> | **Mean ± Std**           | **57.54 ± 0.11** | **30.92 ± 0.10** | **18.17 ± 0.57** | **37.87 ± 0.03** | **60.68 ± 0.20** | **41.00 ± 0.18** | **35.82 ± 0.69** | **47.09 ± 0.07** |
> | **Domain-Specific Data** |                  |                  |                  |                  |                  |                  |                  |                  |
> | Exam         | 58.20            | 33.34            | 22.88            | 38.14            | 60.80            | 40.26            | 35.34            | 45.47            |
> | Math         | 58.15            | 32.48            | 23.50            | 38.04            | 60.88            | 40.25            | 36.43            | 45.85            |
> | Code         | 58.58            | 33.06            | 23.70            | 38.45            | 60.71            | 40.63            | 36.75            | 46.03            |
> | OpenCompass  | 57.67            | 32.07            | 24.60            | 38.11            | 61.16            | 41.65            | 37.78            | 46.86            |
> | **Mean ± Std**           | **58.15 ± 0.38** | **32.74 ± 0.50** | **23.67 ± 0.63** | **38.19 ± 0.15** | **60.89 ± 0.18** | **40.70 ± 0.61** | **36.58 ± 1.03** | **46.05 ± 0.61** |
>
> Key observations:
>
> 1. **Strong generalization**: Even when calibrated on a single domain (e.g., Exam), SERE still maintains reasonable performance on other domains (e.g., Code), demonstrating solid cross-domain robustness.
> 2. **Higher sparsity benefits from domain-specific calibration**: At higher expert skipping rates (K=1), domain-specific calibration consistently yields better performance than general-domain calibration.

---

> ### Author Response · Authors · 2025-11-20
> **Response to Reviewer FvHj [Part 2/4]**
>
> **A1 [Part 2/2]: (2) Comparisons with Static, Data-Free Alternatives**   We explored static similarity calculation alternatives based solely on expert parameters. For each MoE expert with weights $W_{\text{up}}, W_{\text{gate}}, W_{\text{down}}$ (denoted as $\theta_1, \theta_2, \theta_3)$, we evaluated two parameter-combination strategies, following prior work [1]:
>
> - **Concat**: directly concatenate all three weight matrices: $(\theta_1, \theta_2, \theta_3)$;
> - **Logic**: construct a composite weight as $\theta_3 (\theta_1 \cdot \theta_2)$, motivated by the expert forward pass.
>
> We computed similarity matrices using various metrics (Frobenius norm, cosine similarity, linear/RBF/polynomial CKA) on these static representations. Results are shown below:
>
> | Weight Combination | Similarity Metric | K=1            |                |                |                | K=2            |                |                |                |
> | ------------------ | ----------------- | -------------- | -------------- | -------------- | -------------- | -------------- | -------------- | -------------- | -------------- |
> |                    |                   | **Exam**       | **Math**       | **Code**       | **AVG**        | **Exam**       | **Math**       | **Code**       | **AVG**        |
> | Concat             | Frobenius         | 58.41          | 24.09          | 19.54          | 34.01          | 60.69          | 39.01          | 35.72          | 45.14          |
> | Concat             | Cosine            | 58.74          | 25.38          | 13.70          | 32.61          | 60.76          | 39.88          | 34.32          | 44.99          |
> | Concat             | CKA-linear        | 58.24          | 30.20          | 18.54          | 35.66          | 60.76          | 40.06          | 33.82          | 44.88          |
> | Concat             | CKA-RBF           | 58.34          | 30.53          | 19.77          | 36.21          | 60.67          | 39.55          | 32.60          | 44.27          |
> | Concat             | CKA-Poly          | 58.73          | 30.52          | 20.26          | 36.50          | 60.80          | 39.60          | 30.37          | 43.59          |
> | **Mean ± Std**     | **—**             | **58.49±0.21** | **28.14±2.82** | **18.36±2.40** | **35.00±1.47** | **60.74±0.05** | **39.62±0.36** | **33.37±1.80** | **44.57±0.57** |
> | Logic              | Frobenius         | 58.94          | 23.52          | 17.03          | 33.16          | 61.00          | 38.36          | 33.41          | 44.26          |
> | Logic              | Cosine            | 58.89          | 29.19          | 17.74          | 35.27          | 60.61          | 40.43          | 31.69          | 44.24          |
> | Logic              | CKA-linear        | 58.02          | 28.91          | 18.55          | 35.16          | 60.72          | 39.52          | 31.77          | 44.00          |
> | Logic              | CKA-RBF           | 57.76          | 28.55          | 17.13          | 34.48          | 60.57          | 38.24          | 35.34          | 44.72          |
> | Logic              | CKA-Poly          | 58.42          | 28.49          | 16.92          | 34.61          | 60.89          | 40.29          | 32.60          | 44.59          |
> | **Mean ± Std**     | **—**             | **58.41±0.47** | **27.73±2.12** | **17.47±0.61** | **34.54±0.75** | **60.76±0.16** | **39.37±0.93** | **32.96±1.34** | **44.36±0.26** |
>
> These results confirm that static, data-free similarity metrics are viable for SERE to some extent. However, they consistently underperform activation‑based similarity methods, particularly on the Math benchmark, likely because static parameter similarity cannot capture the dynamic behavior introduced by nonlinear activations during expert computation. This highlights the advantage of our activation-driven approach.
>
> The new results have been added in Section 4.4 "ABLATION STUDY" and Appendix A.3 "PARAMETER-BASED SIMILARITY COMPUTATION METHODS" in the revised manuscript.

---

> ### Author Response · Authors · 2025-11-20
> **Response to Reviewer FvHj [Part 3/4]**
>
> **Q2: Focus on Decoding**
>
> **A2:**   Thank you for raising this important point. SERE accelerates decoding by reducing the number of activated experts that lower memory communication overhead, but it does **not** reduce FLOPs. Consequently, it is not expected to accelerate the prefill stage, which is compute-bound.
>
> To validate this, we measured **Time To First Token (TTFT)** under varying query-per-second (QPS) loads. Results are as follows:
>
> | **model / QPS**    | 8     | 16    | 24    | 32     |
> | ------------------ | ----- | ----- | ----- | ------ |
> | Qwen1.5-A2.7B-Chat | 33.64 | 40.62 | 45.33 | 51.43  |
> | SERE K=2           | 33.57 | 38.53 | 45.24 | 50.24  |
> | SERE K=1           | 32.48 | 37.64 | 42.58 | 48.10  |
> | Qwen3-30B-A3B      | 66.72 | 81.08 | 96.02 | 114.03 |
> | SERE K=2           | 65.09 | 78.52 | 92.69 | 104.36 |
> | SERE K=1           | 64.94 | 78.62 | 92.25 | 107.44 |
> | DeepSeekV2-Lite    | 67.06 | 82.57 | 93.12 | 106.44 |
> | SERE K=2           | 66.03 | 79.99 | 91.10 | 108.23 |
> | SERE K=1           | 66.04 | 79.60 | 92.67 | 107.61 |
>
> The TTFT of SERE is nearly identical to the baseline, with minor improvements in some cases, which is consistent with the compute-bound nature of prefill. Moreover, in a typical generation scenario (e.g., 128 input tokens followed by 256 output tokens), prefill accounts for less than 1\% of the total latency, and this proportion will be even smaller when the outputs become longer.
>
> Regarding KV-cache quality: we analyzed the proportion of primary experts retained during prefill across different batched inputs:
>
> | **Model / Batch Config** | **32×128** | **16×64** | **4×256** |
> | ------------------------ | ---------- | --------- | --------- |
> | Qwen1.5-A2.7B-Chat       | 100%       | 100%      | 100%      |
> | Qwen3-30B-A3B            | 94.53%     | 86.71%    | 81.65%    |
> | DeepSeekV2-Lite          | 100%       | 100%      | 100%      |
>
> In most practical batch settings, nearly all experts are retained as primary experts during prefill. Besides, because more experts are active during this stage, re-routing can more reliably identify high-quality substitutes, leading to only a **minimal impact on KV-cache quality**.
>
> To further assess the impact, we implemented and evaluated a hybrid strategy: **apply SERE only during decoding** (i.e., retain all experts during prefill). The results are as follows:
>
> | K    | Method            | Exam      |           |           | Math      |           |           | Code      |           | AVG       |
> | ---- | ----------------- | --------- | --------- | --------- | --------- | --------- | --------- | --------- | --------- | --------- |
> |      |                   | cmmlu     | BoolQ     | bbh       | math      | gsm8k     | math401   | heval     | mbpp      |           |
> | 8    | Baseline          | 84.88     | 90.21     | 76.70     | 72.28     | 89.23     | 79.05     | 87.20     | 78.40     | 82.24     |
> | 2    | Baseline          | 10.01     | 60.52     | 10.48     | 3.38      | 6.97      | 16.96     | 3.66      | 2.40      | 14.30     |
> | 2    | SERE (0.0)        | 81.24     | 89.79     | 71.33     | 70.22     | 82.41     | 80.80     | 82.93     | 63.80     | **77.82**     |
> | 2    | Only Decode (0.0) | 80.31 | 89.42 | 71.81 | 69.60 | 82.41 | 80.80 | 84.15 | 63.60 | 77.33 |
> | 2    | SERE (0.5)        | 81.51     | 90.37     | 74.15     | 72.06     | 85.97     | 81.55     | 85.37     | 72.00     | **80.37**     |
> | 2    | Only Decode (0.5)| 81.65 | 90.12 | 73.50 | 71.22 | 84.38 | 81.05 | 87.20 | 70.20 | 79.78 |
> | 1    | Baseline          | 0.00      | 61.68     | 4.89      | 0.08      | 0.91      | 1.25      | 0.00      | 0.00      | 8.60      |
> | 1    | SERE (0.0)        | 60.53     | 85.08     | 57.64     | 46.98     | 52.08     | 52.12     | 32.32     | 31.40     | **52.27**     |
> | 1    | Only Decode (0.0) | 62.96 | 85.02 | 57.56 | 46.32 | 50.95 | 52.37 | 37.80 | 32.60 | 51.20 |
> | 1    | SERE (0.5)        | 77.89     | 89.76     | 65.45     | 53.40     | 54.28     | 54.86     | 64.02     | 53.20     | **64.11**     |
> | 1    | Only Decode (0.5)| 78.68 | 89.82 | 65.60 | 52.48 | 53.68 | 53.62 | 66.46 | 51.00 | 63.34 |
>
> While "Only Decode" slightly improves performance on some Code tasks, it hurts performance on Math tasks, and **consistently underperforms the original SERE method across different skipping rates and thresholds**. We consider this may be because inconsistent expert selection between prefill and decoding stages introduces a distribution shift.
>
> In summary, although SERE does not provide significant acceleration during the prefill stage, it can be applied safely without degrading KV cache quality or overall performance. The new results have been added in Appendix C.2 "DETAILED ANALYSIS ON PREFILLING STAGE" in the revised manuscript.

---

> ### Author Response · Authors · 2025-11-20
> **Response to Reviewer FvHj [Part 4/4]**
>
> **Q3: In some MoE models, including Qwen3, there is a top-k weight normalization to adjust the total expert weights to 1. Do you keep this operation?**
>
> **A3:**   Yes, we retain this normalization. In fact, SERE is applied **after** top-k weight normalization. As stated in Section 3.2.2 of the manuscript:
>
> > “It should be noted that the re-routing process does not modify the router weights.”
>
> Thus, the sum of expert weights remains 1 after re-routing, preserving consistency with the original MoE formulation.
>
> ------
>
> **Q4: Would this method work for the prefilling stage?**
>
> **A4:**   SERE is specifically designed to accelerate the **decoding** stage by reducing the number of activated experts. Since it does not reduce FLOPs, it **does not accelerate** the prefilling stage in theory. However, as demonstrated in **A2**, SERE provides a slight reduction in prefill latency while maintaining the quality of the initial KV cache. Therefore, SERE is robust and safe to use during prefill.
>
> ------
>
> **Q5: It would be helpful to include standard deviation values to Tables 4 and 5 (AVG column).**
>
> **A5:**   Thank you for this excellent suggestion. We have added **Mean ± Std** annotations to all relevant tables in  Section 4.4 "ABLATION STUDY". The consistently low standard deviations confirm that SERE is robust across different calibration datasets and similarity metrics.
>
> ------
>
> [1] Diversifying the Expert Knowledge for Task-Agnostic Pruning in Sparse Mixture-of-Experts

---

> > ### Comment · Reviewer_FvHj · 2025-11-26
> >
> > I would like to thank the authors for their extensive rebuttal and changes in their revised manuscript which have addressed most of my concerns. I will retain my positive score.

---

### Official Review · Reviewer_gTSr · 2025-11-04

**Soundness:** 4
**Presentation:** 4
**Contribution:** 4
**Rating:** 6
**Confidence:** 4

**Summary:**

This work explores expert re-routing during the inference stage to reduce the number of active experts and thereby improve efficiency. The authors demonstrate a good trade-off between performance and efficiency with their simple yet effective approach.

**Strengths:**

- This work tackles a critical and interesting challenge in MoE serving by employing a simple and intuitive mitigation method.
- The authors' implementation, which features custom CUDA kernels, is good for its easy adaptation to vLLM.
- Backed by several insightful observational experiments, the method achieves surprisingly good results.

**Weaknesses:**

- The method builds on the observation that experts exhibit high similarity scores. Does this phenomenon occur across all MoE models, or is it specifically a byproduct of using upcycling [1] as the initialization strategy?

Minor problem:
- It would be helpful to provide more details about the specific problem this work aims to address. At first glance, readers might assume the paper focuses on improving load balancing. Including key terms such as “expert eviction” could better clarify the intended scope and contribution.

[1] SPARSE UPCYCLING: TRAINING MIXTURE-OF-EXPERTS FROM DENSE CHECKPOINTS

**Questions:**

None

---

> ### Author Response · Authors · 2025-11-20
> **Response to Reviewer gTSr [Part 1/2]**
>
> Dear Reviewer,
>
> We appreciate your thorough evaluation of our paper. Below, we address your concerns and provide our responses, hoping to solve your concerns about the article. All revisions will be included in the forthcoming rebuttal version of the paper.
>
> ------
>
> **Q1: The method builds on the observation that experts exhibit high similarity scores. Does this phenomenon occur across all MoE models, or is it specifically a byproduct of using upcycling [1] as the initialization strategy?**
>
> **A1:** Thank you very much for your question. This is indeed an interesting point. To the best of our knowledge, among the three MoE models used in our experiments:
>
> - The Qwen2 technical report [1, p.5, Section: Expert Initialization] states that **Qwen1.5-MoE-A2.7B** was initialized via upcycling from the dense **Qwen-1.8B** model.
> - The DeepSeek-V2 technical report [2, p.12, Section: Model Hyper-Parameters] indicates that **DeepSeek-V2-Lite** initialized its expert parameters randomly.
> - For **Qwen3-30B-A3B** [3], no public information is currently available regarding its initialization strategy.
>
> Nowadays, most MoE models adopt fine-grained experts, making upcycling from dense models less adaptable. Meanwhile, with the development of training infrastructure and the expansion of training data, training MoE models from scratch has become increasingly feasible. In fact, OLMoE [4] argues that **training from scratch** not only yields better performance than upcycling but also allows greater flexibility in model architecture and hyperparameters.
>
> To further validate the generality of our method, we examined the expert similarity in several other MoE models trained from scratch, including **OLMoE-1B-7B-0125-Instruct** [4], **DeepSeekMoE** [5], and **Ling-mini-2.0** [6]. The visualization results in  Appendix C.3 reveal that all these models contain certain layers where experts are highly similar (e.g., Layer 1 and 5 in Ling-mini; Layer 1, 2, 4, and 15 in OLMoE; Layer 1, 2, 3, and 27 in DeepSeekMoE).
>
> Additionally, we evaluated the acceleration effect of SERE on DeepSeekMoE using vLLM, with the following results:
>
>
>
> |          | Exam  | Math  | Code  | AVG   | TTFT  | TPOT  |
> | -------- | ----- | ----- | ----- | ----- | ----- | ----- |
> | Baseline | 46.11 | 44.48 | 44.51 | 45.03 | 52.54 | 42.74 |
> | SERE K=2 | 44.71 | 42.95 | 44.51 | 44.05 | 52.93 | 39.99 |
> | SERE K=1 | 43.12 | 39.80 | 39.02 | 40.64 | 51.04 | 33.99 |
>
> Taken together, we conclude that **high intra-layer expert similarity is not a byproduct of upcycling-based initialization but rather a common characteristic across MoE models**. In fact, we hypothesize that this phenomenon is more likely caused by load-balancing constraints during training, which enforce functional redundancy among experts to satisfy uniform routing requirements. The new results have been added to Appendix C.3 "SIMILARITY MATRICES VISUALIZATION" in the revised manuscript.

---

> ### Author Response · Authors · 2025-11-20
> **Response to Reviewer gTSr [Part 2/2]**
>
> **Q2: It would be helpful to provide more details about the specific problem this work aims to address. At first glance, readers might assume the paper focuses on improving load balancing. Including key terms such as “expert eviction” could better clarify the intended scope and contribution.**
>
> **A2:** Thank you very much for your insightful suggestion. In the manuscript, we have clearly presented that the core problem our work addresses is **"reducing the number of activated experts during batch decoding in MoEs"**. Specifically:
>
> - Lines 12–15: "However, in production serving, MoE models require batch inference to optimize hardware efficiency, which may cause excessive expert activation and thus slow the memory-bound decoding stage."
> - Lines 42–50: "Despite the theoretical efficiency of MoE architectures, their practical gains are often limited by a mismatch between selective activation and batched inference. In real-world services, multiple user requests are batched to improve hardware utilization. However, tokens within a batch often require different experts, leading to a total number of activated experts far above the per‑token budget."
> - Lines 69–70: "This issue is particularly acute during decoding, where sequential token generation makes the process memory‑bandwidth‑bound."
> - Lines 72–74: "Addressing the conflict between batched inference and sparse expert activation is therefore crucial for unlocking the practical scalability of MoE architectures."
>
> We also explicitly describe our method’s key mechanism as **"re-routing to reduce the number of activated experts"**:
>
> - Lines 15–20: "To tackle this issue, we present SERE, a Similarity-based Expert Re-routing method for Efficient batch decoding in MoE models. SERE dynamically reduces active experts in an input‑aware manner by re-routing tokens from secondary experts to their most similar primary counterparts. It also leverages similarity patterns to identify and preserve critical experts, thereby preventing capability loss."
> - Lines 88–91: "First, many experts within an MoE layer exhibit high functional similarity. Therefore, SERE re‑routes tokens from a subset of experts to their most similar counterparts, reducing the number of active experts with minimal capacity loss."
> - Lines 95–97: "In summary, SERE employs a dynamic, input‑aware strategy that jointly considers token characteristics and inter‑expert similarity, skipping more experts when redundancy is high and fewer when diversity is essential for accuracy."
> - Lines 104–105: "SERE significantly reduces the number of active experts while maintaining model performance, enabling faster decoding."
>
> We believe these descriptions already make our objective, scope, and contribution clear to readers. Besides, throughout the manuscript, we deliberately avoid terms like "expert eviction" and instead use "expert skipping" and "reduce the number of active experts" to describe SERE. This kind of presentation aligns with other work [7] and helps prevent confusion with static expert pruning or merging approaches.
>
> To better address your problem, we have further refined the abstract to clarify the scope and contribution of our work more explicitly. The revised text includes the following additions/modifications:
>
> - Lines 15–16: "To address the fundamental tension between batch decoding and expert sparsity"
> - Lines 17–18: "SERE dynamically reduces **the number of** active experts in an input‑aware manner by re-routing tokens from secondary experts to their most similar primary counterparts."
> - Lines 20–21: "Notably, SERE avoids static expert pruning or merging, instead enabling dynamic expert skipping based on batch-level expert redundancy."
>
> ---
>
> [1] Qwen2 Technical Report
>
> [2] DeepSeek-V2: A Strong, Economical, and Efficient Mixture-of-Experts Language Model
>
> [3] Qwen3 Technical Report
>
> [4] OLMoE: Open Mixture-of-Experts Language Models
>
> [5] DeepSeekMoE: Towards Ultimate Expert Specialization in Mixture-of-Experts Language Models
>
> [6] Every Activation Boosted: Scaling General Reasoner to 1 Trillion Open Language Foundation
>
> [7] Faster MoE LLM Inference for Extremely Large Model

---

> > ### Comment · Reviewer_gTSr · 2025-11-21
> >
> > Thanks authors for the detailed response. My concerns are well addressed. I will remain my original score.

---

### Author Response · Authors · 2025-11-20
**Strengths Summary**

**We appreciate that all reviewers recognized the value of our contributions, including our effective approach to a key MoE serving challenge, our practical CUDA kernel implementation, the significant speedup achievements, and the broad applicability of our method.**

Below, we summarize the strengths of our paper based on the reviewers' evaluations.

## Task Significance

- `gTSr`: "This work tackles a critical and interesting challenge in MoE serving by employing a simple and intuitive mitigation method."

- `FvHj`: "The authors address a relevant problem that limits the efficiency and scalability of MoEs."

## Method Simplicity and Effectiveness

- `gTSr`: "This work tackles a critical and interesting challenge in MoE serving by employing a simple and intuitive mitigation method."

- `FvHj`: "The idea of the method is simple yet meaningful."

- `2FF4`: "The kernel-level optimization and re-routing mechanism are well-aligned with system-level efficiency goals. This is an impressive aspect of the study."

## System Implementation and Engineering Optimization

- `gTSr`: "The authors' implementation, which features custom CUDA kernels, is good for its easy adaptation to vLLM."

- `FvHj`: "The authors provide an efficient kernel pluggable into vLLM."

- `2FF4`: "The kernel-level optimization and re-routing mechanism are well-aligned with system-level efficiency goals. This is an impressive aspect of the study."

## Performance Gains

- `gTSr`: "Backed by several insightful observational experiments, the method achieves surprisingly good results."

- `FvHj`: "The results indicate speedups up to 2x with acceptable accuracy drop."

- `2FF4`: "Extensive experiments on multiple MoE architectures (Qwen, DeepSeek) and benchmarks (OpenCompass, MATH, HumanEval) show up to 2× decoding speedup with minimal accuracy loss (<3%)."

## Insightful Analysis

- `2FF4`: "The analysis in this paper is well-reasoned."

## Clear Presentation and Logical Flow

- `2FF4`: "It begins with a similarity analysis that reveals important insights into the relationships among MoE experts, which then guide subsequent research and design decisions. I like this style."

---

### Author Response · Authors · 2025-12-01
**Rebuttal Summary**

We would like to thank all reviewers for the time and effort they have invested in reviewing our paper. We are grateful for the constructive feedback, which has helped us improve the overall quality of our manuscript. We have carefully considered all the comments and have addressed them all in the revised manuscript as well as the official comment section for each reviewer.  **All reviewers provided positive ratings in the initial review phase. During the rebuttal phase, reviewer `gTSr` and `FvHj` indicated that our responses had addressed their concerns.**

Below, we summarize the main points raised by each reviewer, as well as the revisions and clarifications we have made.

------

## **For Reviewer gTSr**

Main Concerns and Our Actions:

1. **The Generality of Expert Similarity in MoE Models**
   - We tested three additional MoE models (DeepSeekMoE, Ling-mini-2.0, OLMoE) trained from scratch. Results show **expert redundancy exists across all MoE architectures regardless of initialization method**, proving our approach works broadly.

1. **Enhanced Problem Formulation and Terminology Clarification**
   - We revised the Abstract and Introduction to state our goal: **"reducing activated experts during batch decoding"** and our core mechanism: **"re-routing to reduce activated experts"** more clearly.

------

## **For Reviewer FvHj**

Main Concerns and Our Actions:

1. **Limitations and Potential Need for Domain-Specific Calibration**
   - We tested various calibration datasets (general vs. domain-specific). **SERE method maintains robust, strong performance even when calibrated using data from different domains**, addressing concerns about domain-specific requirements.

1. **Comparisons Between Data-Driven and Data-Free Calibration Methods**
   - We have added comprehensive experiments on expert-weight-based similarity matrix construction methods. The results indicate that **these static approaches are inferior to our data-driven calibration methods.**

1. **SERE's Performance During the Prefilling Stage**
   - We have conducted additional experiments during the prefilling stage. The results reveal two key insights:
     - **Enabling SERE has negligible impact on prefilling latency, even slightly reduces Time to First Token (TTFT).**
     - **Enabling SERE has minimal impact on the KV Cache during the prefilling phase.**

1. **Additional Technical Clarifications (Weight Normalization and Standard Deviation Reporting)**

   - Regarding weight normalization concerns, we clarify that **re-routing functions as a post-routing insertion module that only affects expert selection without modifying the original top-k weights.**

   - Regarding SD reporting, **we have added standard deviation values to tables in Section 4.4 for improved statistical rigor.**

------

## **For Reviewer 2FF4**

Main Concerns and Our Actions:

1. **Lack of Theoretical Analysis for Similarity-Based Re-routing**
   - We added a comprehensive theoretical analysis in Appendix A.4 with formal upper bounds on output perturbation under Lipschitz continuity assumptions.

1. **Analysis of Similarity Threshold Interaction with Model Generalization and Stability**
   - We conducted experiments across threshold values 0.0-1.0. Results show **similarity thresholds create optimal trade-off between acceleration and performance**, with reasoning tasks requiring higher thresholds than knowledge tasks.

1. **Domain Sensitivity and Cross-Domain Generalization of Similarity Matrices**
   - We tested calibration on various datasets (general vs. domain-specific data). **SERE maintains highly consistent performance across different calibration sources**.

------

**Final Remarks:**

**We have carefully incorporated all above suggestions and clarifications into the revised version**, including the theoretical analysis, expanded MoE model evaluation, comprehensive domain robustness studies, data-free similarity method comparisons, prefilling stage analysis, detailed similarity threshold analysis, enhanced problem formulation, and improved statistical reporting. We also encourage all reviewers to read our official responses to other reviewers for a more comprehensive understanding of the related issues.

**Once again, our deepest thanks to all reviewers for your generous contributions to this collaborative peer-review process, and we wish you all the best in your future endeavors.**

---

### Meta-Review · Area_Chair_yJ4G · 2025-12-15

**Summary:**

The paper addresses the efficiency bottleneck in MoE models during the batch decoding stage. The authors identify a tension between batch inference (required for hardware efficiency) and sparse expert activation (required for MoE efficiency). They propose SERE (Similarity-based Expert Re-routing), a method that dynamically re-routes tokens from "secondary" experts to functionally similar "primary" experts based on a pre-computed similarity matrix. This reduces the number of active experts per batch. The authors provide a custom CUDA kernel compatible with vLLM and demonstrate up to 2.0x speedups with minimal accuracy loss on various reasoning benchmarks.

**Reviewer Concerns:**

Multiple reviewers shared concerns about the reliance on offline calibration data (potential domain mismatch, comparisons to data-free methods, and sensitivity to domain shifts)

One reviewer questioned if the high expert similarity was an artifact of "upcycling" initialization rather than a general MoE property. Another reviewer felt that the method focused exclusively on decoding, ignoring potential impacts on the prefill stage and KV-cache generation.

**Reviewer Scores:**

The authors provided a comprehensive rebuttal that effectively addressed the major concerns - which I believe would have changed the score if the reviewers had been able to participate fully in the discussion.

- They conducted new experiments on scratch-trained models (DeepSeekMoE, Ling-mini, OLMoE), demonstrating that expert redundancy is a general phenomenon, not limited to upcycled models.

- The authors added experiments with domain-specific calibration sets and compared against data-free static methods (weight similarity). Results showed their activation-based method is superior and robust across domains.

- They provided data showing SERE has negligible impact on prefill latency and maintains high KV-cache quality.

---

### Decision · Program_Chairs · 2026-01-26

Accept (Poster)